# Innovative Galenic Formulation of Prussian Blue Tablets: Advancing Pharmaceutical Applications

**DOI:** 10.3390/ph18101568

**Published:** 2025-10-17

**Authors:** Borja Martínez-Alonso, Guillermo Torrado Durán, Norma S. Torres Pabón, M. Ángeles Peña Fernández

**Affiliations:** Department of Biomedical Sciences, Faculty of Pharmacy, University of Alcalá (UAH), Campus Universitario, Crta. Madrid—Barcelona km. 33.600, Alcalá de Henares, 28771 Madrid, Spain; borja.martineza@uah.es (B.M.-A.); guillermo.torrado@uah.es (G.T.D.); sofia.torres@uah.es (N.S.T.P.)

**Keywords:** Prussian blue, tablet, direct compression, antidote, stability

## Abstract

**Background/Objectives:** Given the persistent threat of war and nuclear accidents, and the global reliance on marketed Prussian blue capsules manufactured in only a few countries without an openly accessible quantitative formulation, there is a critical need for robust tablet alternatives that ensure stability, scalability, and rapid deployment. This study focuses on the design and development of PB tablets for oral administration as decorporation agents for radioactive and toxic species, particularly for treatment in nuclear and radiological emergencies. **Methods:** Advanced tableting processes, including direct compression, wet granulation, and dry granulation, were employed to develop innovative Prussian blue tablet formulations and to provide significant flexibility for industrial-scale production. Comprehensive physicochemical and pharmacotechnical characterizations were performed to support the formulation and to ensure both the safety and efficacy of the PB tablets. Stability studies were conducted in accordance with ICH guidelines to evaluate product performance over time and to confirm that quality and performance attributes remained within specification. **Results:** Among the formulations evaluated, the direct compression (DC5) was recommended for industrial production due to its simplicity, short cycle time, and high throughput. Stability studies up to 18 months confirmed that the PB tablets remained within specification, and the program is ongoing at 24, 36, 48, and 60 months. **Conclusions:** This research provides a promising advancement in countermeasures for nuclear and radiological incidents by delivering a robust, scalable PB tablet formulation that can be rapidly manufactured and deployed in emergency situations.

## 1. Introduction

Prussian blue (PB) is an inorganic metal hexacyanoferrate, often classified as a metal–organic framework (MOF) due to its three-dimensional network of metal–ligand linkages and zeolitic properties. Owing to its structure, PB can retain cations, such as cesium-137 (^137^Cs) [1]. Nuclear power plants are an important source of energy and use the nuclear fission of uranium-235 (^235^U) to generate energy; this process produces various radionuclides such as barium-141 (^141^Ba) and krypton-92 (^92^Kr). One of these fission products is ^137^Cs, which is considered a long-lived fission product. Elements such as iodine-131 (^131^I), with a half-life of 8 days, or strontium-89 (^89^Sr), with a half-life of 50 days, are short-lived radionuclides in contrast to ^137^Cs. ^137^Cs has a physical half-life of 30 years [2,3]. ^137^Cs has a negative effect on ecosystems and human health because of its radioactivity and chemical toxicity. It has high chemical similarity to potassium ions (K^+^); thus, ^137^Cs can be absorbed in the body and cells through a membrane transport mechanism, leading to elemental poisoning and internal radiation exposure. Moreover, due to its high solubility, ^137^Cs can easily accumulate in environmental water; therefore, through infiltration mechanisms, ^137^Cs can enter the soil and then the food chain, thereby affecting animals and humans. A critical attribute of ^137^Cs that contributes to its high radiotoxicity is its predominant emission of high-energy gamma photons (661.7 keV), which possess significant penetrating power and ionization potential. Exposure to these gamma rays can result in severe biological damage because of their ability to ionize atoms and molecules in living tissues [4].

The importance of preventive measures against radioactive or nuclear incidents is evident because of the geopolitical instability of certain regions and the ongoing conflict near Ukraine’s nuclear power plants. Past incidents, such as Chernobyl and the ongoing concerns at Zaporizhzhia, which is Europe’s largest nuclear facility, highlight the potential risks associated with nuclear power. Additionally, decommissioning efforts at plants such as Santa María de Garoña and José Cabrera demonstrate the long-term challenges of nuclear waste management and underscore the need for strict control measures to prevent radiological exposure [5]. The International Atomic Energy Agency (IAEA) classifies nuclear incidents using the International Nuclear Event Scale (INES), which ranges from minor deviations (level 0) to major accidents (level 7). Significant historical incidents include Three Mile Island (level 5), Chernobyl (level 7), and Fukushima (level 7); these incidents required urgent intervention to protect affected populations. In such scenarios, access to effective countermeasures, such as PB, is crucial for mitigating internal contamination and reducing radiotoxicity (Figure 1) [6,7,8,9,10].

In addition to large-scale nuclear accidents, accidental exposure to radioactive materials remains a significant concern, particularly in research and medical facilities. This further reinforces the need to maintain stockpiles of countermeasures such as PB or to ensure the capacity to manufacture and distribute them rapidly when needed [11].

Antidotes are among the most crucial countermeasures in medical practice because once a toxin is identified, an effective antidote is needed to reverse its effects and promote patient recovery. In the context of radioactive exposure resulting from nuclear fission events, ^137^Cs is one of the most abundant and harmful isotopes. Currently, the only available pharmaceutical product in Europe is Radiogardase^®^, which is marketed as 500 mg PB hard capsules. Its SmPC discloses the qualitative composition. However, the quantitative composition of excipients and the exact manufacturing process are not publicly available [12]. Moreover, production is concentrated in a limited number of countries, which can constrain global accessibility and highlights the need for alternative dosage forms. PB, or iron (III) hexacyanoferrate, is an insoluble compound that is not absorbed through the gastrointestinal mucosa following oral administration. In the gastrointestinal tract, it acts as a chelator that binds and sequesters monovalent cations with a high affinity for ions with a relatively large ionic radius, particularly ^137^Cs, and, to a lesser extent, thallium. Moreover, because cesium undergoes enterohepatic recirculation, the binding of PB with ^137^Cs in the intestine following biliary excretion prevents its reabsorption; thus, this process interrupts the ^137^Cs cycle and facilitates its elimination via feces, ultimately reducing the duration of internal exposure to this radioisotope [13].

Under standard treatment protocols for acute radioactive exposure, adults and children over 12 years of age are administered six capsules three times daily, whereas children between 2 and 12 years of age receive two capsules three times daily. These capsules are intended to be swallowed whole; however, in patients with swallowing difficulties, the contents may be mixed with food or liquids. Additionally, treatment with PB needs to begin as soon as possible, and the recommended duration is at least 30 days; the duration depends on the severity of exposure and clinical judgment. Once internal radioactivity has significantly decreased, the dosage can be reduced to 1–2 g three times daily to improve gastrointestinal tolerance. Clinical trials have demonstrated that the oral administration of 3 g of PB per day reduces the biological half-lives of ^137^Cs from 110 days to 40 days and thallium from 8 days to approximately 3 days [14].

Beyond the approved capsule product Radiogardase^®^, several groups have explored alternative formats of PB to enhance handling and decorporation performance. Examples include porous PB-cellulose aerogels specifically designed for gastrointestinal decorporation [15], PB-embedded magnetic hydrogel beads for efficient cesium removal [16] and a broader range of PB-based nanomaterials for biomedical applications [17,18,19]. From a formulation standpoint, a recent oral PB product with pH-modifying excipients sought to improve thallium binding under gastric conditions, illustrating a “Radiogardase-like” optimization strategy rather than a new solid form [20]. Collectively, these studies demonstrate active development around PB delivery; however, to our knowledge no peer-reviewed work has reported PB tablets produced by direct compression, which contextualize the novelty and significance of the present study.

Owing to the critical importance of antidotes and the limited global manufacturing footprint of available PB products and given that only the qualitative composition of Radiogardase^®^ is publicly disclosed in the SmPC (whereas quantitative composition and process details are not), the AZPINDECS project was initiated by the Spanish Ministry of Defense. Within this framework, a strategic decision was made to develop a military-grade CBRN countermeasure in the form of tablets containing 500 mg of PB as the active pharmaceutical ingredient (API). A tablet formulation is particularly advantageous in emergency scenarios: direct compression offers shorter production cycles and higher yields than capsule production, with fewer processing stages and facility requirements. The PB used in this study for tablet development was locally synthesized at the University of Alcalá; additionally, the physicochemical and pharmacotechnical characterization of this PB, as well as its Cs-binding capacity, was detailed in the publication “Physicochemical and pharmacotechnical characterization of PB for future PB oral dosage forms formulation” [21].

A primary goal of this project was to develop a pharmaceutical form that is both easy and rapid to produce, and this goal was achieved using the direct compression manufacturing technique. This method has significant advantages over traditional hard gelatin capsules; these advantages include a shorter production cycle and reducing the reliance on specialized components (e.g., capsule shells), although other excipients are still procured from qualified external suppliers; these factors are critical considerations for emergency medicines. By incorporating excipients already used in other medications from the Military Pharmacy Center of Defense in Spain and selecting widely available excipients from the pharmaceutical industry, this approach minimizes potential supply chain issues in crisis situations [22,23].

The primary objective of this paper is to report the formulation, manufacture, and characterization of PB tablets; these tablets were developed not only to meet the requirements of the AZPINDECS project of the Ministry of Defense, but also to provide an open-access, reproducible formulation that can be rapidly produced and deployed in emergencies. In addition, we present a comparative study evaluating the cesium-binding efficacy of newly developed PB tablets against the marketed drug Radiogardase^®^ to validate the effectiveness of the new formulation. Finally, a comprehensive stability study under accelerated conditions and data from the first 18 months of the stability testing under Zone II climatic conditions in accordance with the International Council for Harmonization of Technical Requirements for Pharmaceuticals for Human Use (ICH) guidelines [24] are included.

## 2. Results and Discussion

### 2.1. Excipient Selection

All excipients proposed for the tablet formulations were chosen after an extensive literature review, and special attention was given to avoid API-excipient incompatibilities. Microcrystalline cellulose is widely used as a diluent in tablet formulations for direct compression as well as wet or dry granulation. It also possesses glidant properties. It is stable but slightly hygroscopic; therefore, it requires humidity-controlled storage. It is also considered non-toxic and non-irritant [25]. Another widely used diluent for compression processes is mannitol. Owing to its granular morphology and spray-dried production, this excipient exhibits excellent properties for direct compression and dry granulation and low hygroscopicity, which results in a relatively high drying speed; this characteristic is ideal for wet granulation. One of its most notable disadvantages is that it reduces tablet hardness when used. Both diluents were used in the proposed formulations; however, mannitol was discontinued due to insufficient hardness in direct-compression tablets, and the removed fraction was replaced with microcrystalline cellulose [26,27]. Polyvinylpyrrolidone K30 (povidone K30) is a spray dried grade povidone that yields near-spherical particles with good flow and compressibility. Higher K-value grades (≥60) generally show higher viscosity and less favorable flow for direct compression. In tablets, povidone is used mainly as a binder (and can contribute to disintegration when used intragranularly). Because povidone is water-soluble, it is an excellent binder for wet granulation; accordingly, it was included in all formulations, regardless of the manufacturing process [28]. To improve wet granulation, corn starch was added to the wet granulation formulation (WG). Corn starch is a classical binder commonly used in wet granulation processes. It is used for its water uptake and swelling capacity, which, in conjunction with povidone K30, ensures adequate wetting during wet granulation with water [29]. To improve tablet hardness and avoid the occurrence of capping, a low level of hydroxypropyl methylcellulose was included in all direct compression formulations. This excipient is more commonly used as a matrix former for controlled-release systems at 10–80% (*w*/*w*); however, at 0.25–5% (*w*/*w*) it can function as a binder and help prevent capping [30].

PB acts locally in the gastrointestinal tract; therefore, rapid tablet disintegration after oral administration must be ensured. In tablets, this disintegration could be compromised by the forces employed in dry granulation process, in which the excipient mixture is compacted by applying pressing, and pellets that are partially broken are generated as the dry granules for compression. This brakeage could lead to small particles that are good for compression but could have disintegration difficulties. To prevent this issue, an excipient with good disintegration ability is added to the dry granulation formula. Sodium croscarmellose is used as a disintegrant in tablets at 0.5–5%. It could be used in direct compression processes and was initially considered for use in the direct compression formulations; however, owing to the lack of hardness in the initial formulations (DC1, DC2 and DC3) and the good disintegration times observed in DC4 and DC5, sodium croscarmellose was not included in the final direct compression formulation [31,32].

To ensure required flow properties, lubricants are commonly required in tablet formulations to reduce friction and sticking during compression. The most used lubricant is magnesium stearate, which is used in tablets ranging from 0.25% to 5%. It was initially included in DC2 and DC3; however, owing to a better performance using glyceryl behenate in the compression blend and potential laxative effect of magnesium stearate found in the literature, magnesium stearate was excluded this study [33].

Although glyceryl behenate is used as an excipient for direct compression with lubricant properties that are not as extended as those of magnesium stearate, it can be used in tablets as a lubricant agent in 0.5% to 3% amount. PB API exhibited good flowability in prior studies [21]. The mixture, in which PB represents a 71.43% of the formulation has a good flowability because the flowability of PB is good in itself. Therefore, the addition of glyceryl behenate as a lubricant is sufficient to reach the flowability requirements for direct compression manufacturing. In addition, glyceryl behenate does not affect tablet hardness and is not affected by segregation in the blending procedures. For these reasons, glyceryl behenate was included in direct compression formulations [34].

One requirement that all excipients must meet is as follows: they must be immediately available in local markets for emergency use. PB tablets are a countermeasure and are designed to be a treatment for cases of accidental or intentional exposure to radioactive Cs or Tl. Therefore, to ensure the availability of all formula components in case of need, all excipients selected are frequently used and easily accessible, and more innovative excipients with difficult market access are discarded.

To summarize, all excipients selected for the formulations are well established and commonly used because of their good performance in direct compression formulations, with corn starch used exclusively in the WG. The final selection comprised those excipients that provided the best tableting properties.

### 2.2. Tested Formulations

Table 1, Table 2 and Table 3 present the developed formulations for direct compression and for wet granulation or dry granulation processes, respectively. Additionally, the WG required 30% (*w*/*w*) Milli-Q water to achieve adequate wetting for granule formation.

### 2.3. Formula Components Compatibility Study Using Differential Scanning Calorimetry (DSC) and Fourier Transform Infrared Spectroscopy (FTIR)

Figure 2 shows the thermograms of the raw PB material and all binary PB-excipient combinations that were obtained. For completeness, DSC thermograms of the individual components (PB and each excipient) were also acquired under identical conditions; the corresponding reference profiles are provided in the Appendix A and are cited, where pertinent, to support the assignment of thermal events observed in the PB-excipient binary systems shown in Figure 2. The compression blend of the final selected tablet formulation, DC5, was also analyzed under the same conditions, and the rationale for selecting this formulation and the corresponding compression methodology are discussed in detail later in this paper. The DSC analysis of the raw PB material revealed an endothermic event from 158 °C to 205 °C corresponding to the water contained in its crystalline structure, which can be released at low or moderate temperatures [21]. Water loss can also occur over a broader range, approximately 150–300 °C, depending on the quantity and type of retained water.

Microcrystalline cellulose exhibits pronounced hygroscopicity and characteristically displays a broad, shallow endothermic event at 50–120 °C consistent with bound-water loss. This endothermic event is evident in the raw material analysis (Appendix A, provided in the Appendix A), but not in the binary sample with PB 1_1, due to its low intensity, it cannot be appreciated in comparison with the endotherm corresponding to the PB peak. The glass transition for hydroxypropyl methylcellulose usually occurs in the range of 120–190 °C and depends on factors such as molecular weight and the degree of substitution of the hydroxypropyl and methyl groups. Povidone K30 is an amorphous polymer that does not exhibit a defined melting point and has a glass transition in the range of 140–180 °C. Croscarmellose sodium is a partially amorphous polymer, and DSC reveals a glass transition in the range of 150–180 °C (Appendix A, provided in the Appendix A). In the binary mixtures with PB, the strong endothermic event of PB, observed between roughly 160 and 205 °C, overlaps and masks these weaker glass-transition signals of the excipients. Nevertheless, the thermograms of PB-microcrystalline cellulose, PB-hydroxypropyl methylcellulose, PB-povidone K30, and PB-croscarmellose sodium exhibit endothermic peaks from 163.12 °C to 205.0, 152.6 °C to 204.3 °C, 164.8 °C to 205.4 °C, and 135.3 °C to 210.0 °C, respectively. No new peaks or disappearance of characteristic events were observed, indicating compatibility with the four evaluated excipients.

Mannitol typically exhibits a sharp melting endotherm in the 165–169 °C range. In the PB-mannitol thermogram the broad PB endotherm that begins near 155 °C partly overlaps the mannitol melting peak. After repeating the binary analysis and comparing it with the thermogram of the pure mannitol raw material (Appendix A, provided in the Appendix A), the presence of the mannitol melting event was confirmed, appearing as a shoulder on the main PB endotherm. This overlap reflects simple thermal superposition, with no evidence of chemical interaction between PB and mannitol. The PB-corn starch thermogram reveals a broad endothermic event from 121.2 °C to 210.2 °C. Although broad, corn starch most likely lowers the energy needed for PB melting and water loss because of its semicrystalline nature [35]. The combination of this PB shift with the starch transition (120–160 °C) likely explains the broad band and does not indicate instability.

Although small increases or decreases in PB transition temperatures may occur due to the excipient’s nature and thermic transitions, none of the above-mentioned PB-excipient binary blends indicates an incompatible interaction between PB and the corresponding excipient. Therefore, in addition to the bibliographic information collected, these excipients are considered compatible with PB for their combined use in the manufacturing of the tablets.

Glyceryl behenate has a melting temperature in the range of 70–85 °C. In the updated PB-glyceryl behenate thermogram, the characteristic PB endotherm (≈155–200 °C) is now clearly visible together with the lipid-melting peak. The slight downward shift in the lipid transition is likely due to sample packing and does not signify chemical incompatibility; similar behavior has been described in the literature [36].

In the PB–magnesium stearate blend, a small endothermic event is seen around 100–125 °C followed by a weak exothermic signal near 135 °C and a further endotherm at about 240 °C. The PB endotherm appears broadened and partially degraded compared with the pure API, suggesting a possible interaction between PB and magnesium stearate. Because this interaction could compromise PB quality and, as noted earlier, the formulations containing magnesium stearate also showed the poorest flow properties, magnesium stearate was not selected as an excipient for the final PB tablet formulation. Finally, in the DC5 compression blend, a lower-temperature PB transition zone can be observed without any other deformation in the range of temperatures from 138.2 °C to 202.1 °C. No other peak or event is observed in the thermogram; these results indicate that PB and the excipients chosen for the final formulation DC5 are compatible with the amounts specified for the formulation and suitable for their use in the manufacturing of PB tablets.

FTIR analysis was used as a complementary tool to DSC to detect possible chemical interactions between PB and the excipients of the DC5 formulation. The infrared spectra of PB showed a broad band around 3400 cm^−1^, characteristic of O–H stretching of crystallized or zeolitic water, and an absorption near 3630 cm^−1^ attributed to coordinated water. The intense band near 2000 cm^−1^ corresponds to the C≡N stretching of the metal-coordinated cyanide groups of PB. These characteristic PB bands were preserved in all binary mixtures and in the complete DC5 formulation (Figure 3) [21]. Additional bands observed in DC5 at ~2900 cm^−1^ correspond to C–H stretching vibrations from cellulose derivatives and glyceryl behenate, whereas the broad signal near 3450 cm^−1^ arises from O–H stretching of the cellulose matrix, with a possible minor contribution from the pyrrolidone ring of povidone. No significant shifts, disappearance, or formation of new peaks were detected when comparing the spectra of the mixtures or the DC5 blend with those of the individual components, indicating no detectable chemical interactions between PB and the excipients (Appendix A, provided in the Appendix A).

### 2.4. Intermediate Product Characterization

All compression blends were manufactured, and prior to compression for tablet development, their angle of repose, flow time, Hausner ratio (IH) and Carr index (IC) were determined. Table 4 lists the analytical results for the DC1, DC2, DC3, DC4, DC5, WG and DG formulations.

The values in Table 5 were used to evaluate the results obtained for the angle of repose, IH, and IC and to establish the compressibility and flow characteristics of the formulations Given the crystalline nature and high proportion of PB in all blends, flow times are remarkably low, i.e., under 2 s; consequently, all samples flowed readily through the funnel, predicting good feed behavior in the tablet press during manufacturing [21,37].

The results shown in Table 4 and Table 5 show that all formulations have good flow and compressibility properties, indicating good behavior in the compression steps for all formulations. A deeper analysis confirms that the formulations with magnesium stearate (DC2 and DC3) have worse flow properties and compressibility than the formulations that use glyceryl behenate. These results are reinforced by the information obtained during the bibliographic review, and an explanation is provided in the Excipient Selection section. Therefore, based on these data, it was decided to choose glyceryl behenate as a lubricant. An increase in glyceryl behenate content from 0.5% to 1.0% slightly improved the flow properties of DC4, but subsequent analysis of the resulting tablets revealed that an improvement in compressibility was still needed to avoid tablet breakage or defects. To achieve this goal, mannitol, which, as explained previously, can decrease tablet hardness, was removed and replaced with microcrystalline cellulose in DC5. This change improved the intermediate product compressibility, and the resulting tablets showed no breakage or defects attributable to poor compression of the blend. Therefore, DC5 was chosen as the final formulation for direct compression. As the final formulation for direct compression, DC5 was further evaluated using the sediment delivery model (SeDeM); results are presented in the SeDeM subsection.

The values for WG indicate both good compressibility and good flow behavior, as expected after wet granulation and a screening process. Moreover, the wetting step with water is not affecting the PB raw material because of its insolubility, and its mechanism of action remains unaltered because of the need for water in the PB structure to capture Cs and Tl molecules [21]. The DG formulation exhibited slightly worse compressibility and flow properties but was still acceptable under the defined criteria. This difference between the granules obtained from wet and dry granulation processes is related mainly to the difference in granule shape. Wet granulation leads to more uniform and spherical shapes, which flow better and are more compactable; dry granulation produces irregularly shaped granules, which results in more intergranular voids during compression and compaction, and poorer flow capacity [38].

### 2.5. Tablets Characterization and Final Formulation Selection

#### 2.5.1. Physical and Physicochemical Characterization and Choice of the Final Formula

Once manufactured, tablets of each formulation were characterized; their shape was examined using visual description, and their weight, dimensions, disintegration time, hardness and friability were recorded. Table 6 summarizes the results for the tablet description, weight and dimensions.

First, in the tablets obtained for the formulations for direct compression of DC1, DC2, DC3 and DC4, some defects, such as cracks, tears, scratches, and missing fragments, were observed in the tablets. DC2 and DC3 had a noticeably greater number of these defects than DC1 and DC4, and only a few tablets with the DC4 formulation had defects. Consistent with these visual observations, the mechanical results in Table 6 show insufficient robustness for these four blends: DC2 and DC3 exhibited the lowest breaking strength and failed the friability test, whereas DC1, despite slightly higher breaking strength, also failed due to cracking; DC4 performed comparatively better within this group but still did not meet the pharmacopeial friability limit. The main causes of these defects were the low compressibility of these formulations and the presence of components that reduced the breaking strength and consistency of the tablets; these components included mannitol for all 4 of these formulations and magnesium stearate for DC2 and DC3, as reflected in their lower mechanical strength and failure to meet the pharmacopeial friability limit (Table 6). The insufficient flowability of magnesium stearate resulted in insufficient die filling during the compression process and produced lighter tablets with greater variation. In addition, reducing the structural resistance with both magnesium stearate and mannitol facilitated the occurrence of damaged tablets. Due to the presence of these defects, the formulations needed to be modified. For this reason, magnesium stearate and mannitol were eliminated from the formulations, mannitol was replaced with the previously used diluent, microcrystalline cellulose, and glyceryl behenate was used as the only lubricant. Consistently, DC5 exhibited improved mechanical properties (hardness 41.3 N; friability 0.82%), complying with the usual < 1% friability specification and aligning with the absence of visual defects. Notably, the hardness targeted for DC5 was deliberately moderate, providing sufficient mechanical integrity (with friability within specification) while preserving the very rapid disintegration required for the intended use. This variation in the DC5 formulation led to successful production of tablets without visual defects and a more consistent weight (overall yield > 95% on development batch, later the stability batches had a superior overall yield of ≈98%). The improvements in IH and IC shown in Table 4 support the results summarized in Table 5; here, the tablets were more accurate with respect to the nominal tablet weight of 700.0 mg and had a greater mechanical strength because of the better compressibility and compactness of the DC5 formulation.

In addition, the high disintegration achieved by all formulations was notable. The rapid disintegration of these tablets was crucial for their intended use in treating Tl or radioactive Cs poisoning since it ensured faster onset and enhanced efficacy of the API, PB [39,40,41,42].

Based on all these findings, since the DC5 formulation produced tablets that satisfied all desired properties, this formulation was selected as the definitive formulation for tablets obtained by direct compression.

For the tablets obtained by wet granulation, the WG formulation also showed good performance during the compression process and resulted in tablets with good characteristics and no defects (overall yield ≈ 90%, for detailed step-by-step yields, consult Appendix A, provided in the Appendix A). The results revealed that a very close match to the nominal weight was achieved, resulting in even more compact tablets that continued to effectively disintegrate. Consistently, the mechanical results in Table 6 indicate adequate robustness for WG, combining high breaking strength with low friability while preserving the very rapid disintegration required for the intended use, achieving a shorter disintegration time than Radiogardase^®^. Contrary to expectations, the intermediate product of the DG formulation demonstrated favorable results and produced tablets that appeared to meet the required specifications. However, subsequent analysis revealed that the tablets were not correctly compacted or compressed. This result was shown by the attainment of greater height of the tablets than expected. Furthermore, the mean weight of the tablets was lower than those of the other formulations. The combination of these two factors led to more fragile tablets. This interpretation is in line with the mechanical data in Table 6, which point to reduced breaking strength, despite friability remaining within specification, and thus to inadequate compaction of DG. The cause of these defects was most likely the irregular shape detailed in the previous section, which was typical of the granulation method. Owing to limitations in the available dry granulation process (overall yield ≈ 70%, for detailed step-by-step yields, consult Appendix A, provided in the Appendix A), further optimization of this method was not pursued within the scope of the current study.

The results obtained provided three viable formulations, one for each of the most widespread tablet manufacturing methods: direct compression and compression of granules obtained by wet or dry granulation.

Since the investigation was associated with the AZPINDECS project, one of the formulations needed to be selected for the efficacy tests and stability studies. Two kinds of factors were considered to aid in this decision. The first was the tablet performance on the examined characteristics, and the second was aligned with the priorities and needs established by Spain’s Ministry of Defense.

Considering the first factor, the DG formulation was discarded because its properties were significantly inferior to those of the DC5 and WG formulations. DC5 and WG tablets were more weight accurate and performed more effectively in the compression process; additionally, the characteristics of their intermediate products were better. To decide between DC5 and WG, the second factor was applied. PB is a drug, and its use is associated with the accidental or intentional poisoning with radioactive Cs and Tl; thus, access to this medicine needs to be readily available. For this reason, the direct compression method is preferable to the wet granulation procedure. Direct compression is faster since it involves fewer steps in production, with no need for granulation, screening or drying processes. Furthermore, the absence of the need to use water in direct compression enables the manufacture of these tablets in water-limited environments. Therefore, the final tablet formulation for the efficacy and stability studies was DC5 (complete finished product specifications can be found on Appendix A provided in the Appendix A).

The appearance of the selected final tablets is shown in Figure 4.

#### 2.5.2. Assay and Content Uniformity

Using the validated ICP-OES method (Section 3.8.3), the PB tablets showed an assay of 101.3% of the label claim at release (SD = 0.65%, *n* = 10). Under long-term Zone II storage conditions, the 18-month time point gave 97.3%, also within the predefined acceptance range.

The mean content of the 500 mg PB tablets was 101.5% of the label claim, with a standard deviation of 0.72%, giving an acceptance value (AV) of 1.44, which is well below the Ph. Eur. limit of 15.0. Therefore, the batch fully complies with the requirements of Ph. Eur. 2.9.40 for content uniformity, and no additional 30-unit testing was required.

#### 2.5.3. Dispersion Test

Figure 5 compares the dispersion profiles of the newly developed PB tablets with those of Radiogardase^®^. PB tablets reached ≈100% dispersion within 15 min (mean 99.9 ± 3.5%), whereas Radiogardase^®^ required 30 min to achieve a similar extent of dispersion (mean 99.7 ± 6.5% at 30 min). At 5 min, PB tablets already dispersed to 70.6 ± 1.5%, compared with 36.7 ± 12.0% for Radiogardase^®^.

These data demonstrate that the new PB tablets disperse more rapidly and completely than the reference product, which may favor an earlier onset of decorporation activity in vivo and support the robustness of the formulation.

### 2.6. SeDeM Methodology Study

Since the final selected formulation was a direct-compression formulation, a SeDeM study was carried out on the DC5 compression blend. The SeDeM method was used to determine the suitability of the DC5 blend for direct compression. The pharmacotechnical parameters included the following: the bulk density (rDa) and tapped density (rDc) (dimensional incidence factors); interparticle porosity (rIe), Carr index (IC) and cohesion index (rIcd) (compressibility incidence factors); Hausner ratio (IH), angle of repose (ra) and flow time (rt’’) (flowability incidence factors); loss on drying (rHR) and hygroscopicity (r%H) (lubrication/stability incidence factors); and finally, particles < 50 µm (r%Pf) and homogeneity index (rIθ) (lubrication/dosage incidence factors) (Table 7); these parameters were experimentally determined and mathematically processed to express them in a graphical representation in the form of an SeDeM diagram (Figure 6).

According to the results from the SeDeM analysis, the DC5 formulation is composed of PB, microcrystalline cellulose, povidone K30, hydroxypropyl methylcellulose and glyceryl behenate and is considered suitable for use as a direct compression formulation; these results support the previously obtained data in the characterization of the intermediate product and the manufactured tablets. The acceptance indices are above 0.50 for the parametric index and above 5.00 for the parametric profile index and good compression index; thus, the DC5 formulation is adequate for direct compression. These results are further supported by Da and Dc of 0.66 g/mL and 0.74 g/mL, respectively; both exceed 0.50 g/mL. The flowability of the formulation is good, with an incidence factor of 7.51, and ensures the appropriate behavior of the formulation during manufacturing. The loss on drying is high but necessary, as PB’s structural water is required for its intended effect. Considering its very low hygroscopicity (0.48%), good homogeneity index, and final parameter values, the DC5 formulation is confirmed to be suitable for direct compression.

### 2.7. Cs Binding Capacity of the Tablets

The ability of the PB tablets (DC5 formulation) to bind Cs was determined via an in vitro assay as a surrogate measure of the PB decorporant capacity for radioactive Cs and Tl. For reference, the same assay was carried out with Radiogardase^®^-Cs 500 mg hard capsules, the only marketed medicinal product that uses PB as the active ingredient. The PB tablets bound from 171 to 181 mg Cs/g PB corresponding to 89.0 ± 1.7% of Cs available. Radiogardase^®^ capsules showed Cs binding ranged from 170 to 185 mg Cs/g PB; corresponding to 86.5 ± 2.1% (Figure 7). These results indicate comparable Cs decorporation capacity between the PB tablets and Radiogardase^®^; thus, the PB tablets developed in this study could be used as effective antidotes for treating Cs poisoning.

### 2.8. Stability

Three batches of PB tablets were prepared according to the DC5 formulation. These tablets were packaged in PVC/PVDC-aluminum blisters. Samples from the three batches were stored in climatic chambers under defined temperature and humidity conditions in accordance with ICH guidelines [17] (see Section 3). To date, 18 months under climatic Zone II long-term conditions and 6 months under accelerated conditions have been completed. The results presented in Table 8, Table 9 and Table 10 are reported as mean ± SD across the three batches, with low batch-to-batch variability. The PB tablets were successfully evaluated under both accelerated conditions and long-term (Zone II) conditions up to 18 months. No significant reductions in Cs binding capacity or water content were observed. These results ensure that the efficacy of PB as a radioactive Cs and Tl decorporant is maintained during storage. Moreover, no defects were observed in any tablets during stability tests. Although hardness and friability showed slight decreases, all values remained within the specification limits and the associated standard deviations were small, and weight variation was minimal with no trends; thus, the formulation and compression procedures are adequate for the manufacturing of durable PB tablets. Furthermore, the successful completion of the stability tests indicates that the selected packaging material, PVC/PVDC-aluminum blisters, in conjunction with the inherent properties of the tablets, effectively preserves the product’s characteristics throughout the evaluated stability period.

PB tablets also passed the microbiological tests. These results were expected because of the inorganic nature of PB and its high proportion (71.43%) in the formula. Since the stability studies are satisfactory at 18 months, the stability studies will be extended as planned, and the tablets will be tested again at 24 months. If compliant, testing will proceed at 36, 48 and 60 months. Sixty months is the maximum legally allowed shelf life for tablets in Spain [37].

## 3. Materials and Methods

### 3.1. Active Pharmaceutical Ingredient (API)

In this study, PB used was previously synthesized by the Center for Applied Chemistry and Biotechnology of the University of Alcalá. The PB batch used for the manufacture was DFO-2009-083 (Sigma-Aldrich, St. Louis, MO, USA). Characterization of this batch is fully described in the article “Physicochemical and pharmacotechnical characterization of PB for future PB oral dosage forms formulation” [14].

Radiogardase^®^-Cs (Haupt Pharma Berlin GmbH, Berlin, Germany) 500 mg hard capsules (batch No. 24002125) were used as a reference for comparison.

### 3.2. Excipients

Seven formulations were prepared using the following raw materials as excipients in the proportions specified in Table 1, Table 2 and Table 3.

Milli-Q water was obtained from a Milli-Q water purification system (Billerica, MA, USA). The microcrystalline cellulose was extra pure (Acros Organics, Geel, Belgium, CAS No. 9004-34-6). Mannitol (Mannogen^®^ SPI PharmaTM, New Castle, DE, USA, CAS No. 69-65-8), polyvinylpyrrolidone MW 50 000 K-30 (Acros Organics, CAS No. 9003-39-8), sodium croscarmellose (JRS Pharma, Rosenberg, Germany, CAS No. 9004-32-4), glyceryl behenate (Compritol^®^ 888 ATO, Gattefossé, Saint-Priest, France, CAS No. 91052-55-0), magnesium stearate (Guinama S.L.U. La Pobla de Vallbona, Spain, CAS No. 91031-63-9), corn starch (Guinama S.L.U. La Pobla de Vallbona, Spain, CAS No. 9005-25-8), and hydroxypropyl methylcellulose (Guinama S.L.U. La Pobla de Vallbona, Spain, CAS No. 9004-65-3) were used. All products met pharmaceutical-grade standards.

### 3.3. Drug-Excipient Compatibility

PB and excipients raw material, all possible PB-excipient binary blends, and the final direct compression formulation were characterized using differential scanning calorimetry (DSC, TA DSC25, TA Instruments, New Castle, Estados Unidos, DE, USA). The temperature calibration of the instrument was performed with the Indium Calibration Reference Standard. The samples were weighed into aluminum pans and hermetically sealed with aluminum lids. Each sample was subjected to a heating ramp from 25 to 250 °C at a heating rate of 10 °C/min under a constant nitrogen flow of 50 mL/min. An empty pan subjected to the same conditions was used as the reference.

To reinforce the study of the compatibility between PB and the excipients in the selected DC5 formulation, an FTIR analysis was performed. Infrared spectra were recorded on a PerkinElmer^®^ System 2000 FT-IR spectrometer (PerkinElmer, Waltham, MA, USA). Approximately 2.5 mg of each test sample (pure PB, each individual excipient, PB–excipient binary mixtures, and the final DC5 formulation) were accurately weighed and thoroughly mixed with spectroscopic-grade potassium bromide (KBr, Panreac Química S.L.U., Barcelona, Spain) to a final weight of 250 mg. The mixtures were ground in an agate mortar to ensure a homogeneous fine powder and then pressed into KBr pellets by applying a force of 10 tons for 15 s.

Each spectrum was acquired by averaging ten scans in the range of 4000–400 cm^−1^ at a resolution of 4 cm^−1^. The characteristic absorption bands of PB and possible shifts or new peaks were compared among the pure components, the binary mixtures and the complete DC5 formulation to detect any chemical interaction.

### 3.4. Blending

#### 3.4.1. Direct Compression

As a first step in tablet development, all components listed in Table 1, Table 2 and Table 3 were weighed using a Kern Kb 360-3N precision weighing scale (Kern & Sohn GmbH, Balingen, Germany). For the direct-compression formulations (DC1 to 5), all ingredients, except for the glidants (magnesium stearate and/or glyceryl behenate), were blended for 5 min at 40 rpm in a PP Sabadell OX type number 41411 V-type blender (P. Prat S.A., Sabadell, Spain). Once this blending was complete, the corresponding glidants were added to the initial mixture and further blended for 2 min at 60 rpm.

#### 3.4.2. Wet Granulation

For WG, the API and excipients were blended as follows: first, PB, microcrystalline cellulose, povidone K30, and corn starch were mixed for 5 min at 40 rpm. This blend was then granulated and dried as described in the wet granulation subsection. Finally, the resulting granules were blended with glyceryl behenate for 2 min at 60 rpm.

#### 3.4.3. Dry Granulation

For dry WG, the blending process was carried out in the following steps. First, PB, microcrystalline cellulose, povidone K30, and half of the total amount of croscarmellose sodium were blended for 5 min at 40 rpm. The blend was then granulated using the dry granulation method, as detailed in the dry granulation subsection. Finally, the granules were blended with glyceryl behenate and the remaining croscarmellose sodium for 2 min at 60 rpm.

### 3.5. Granulation

#### 3.5.1. Wet Granulation

Wet granulation was performed in a laboratory double-Z kneader (Ancimo S.L., Barcelona, Spain) operated at 60 rpm for 10 min. A dry blend of PB, microcrystalline cellulose, povidone K30 and corn starch was processed while Milli-Q water was added gradually at an approximate rate of 10% of the total required volume per minute, until reaching the endpoint defined as a cohesive wet mass forming agglomerates without free liquid. In total, 30% (*w*/*w*) Milli-Q water relative to the dry blend was added; for the 1.00 kg development batch, this corresponded to 300 g of water. The wet mass was then passed through an oscillating granulator (Ancimo S.L., Barcelona, Spain) fitted with a 1.0 mm mesh, operating at 40 rpm, to obtain wet granules. The granules were dried in a tray oven (Selecta Furnace model 207, J.P. Selecta, Spain) at 90 °C using a two-shelf configuration. Granules were spread in stainless-steel trays in a shallow layer, and moisture was checked periodically with a Mettler Toledo^®^ LJ-16 moisture analyzer until a residual moisture of ≤1% (*w*/*w*) was reached, with a total drying time of approximately 1 h.

#### 3.5.2. Dry Granulation

For dry granulation, the initial blend of PB, microcrystalline cellulose, povidone K30 and sodium croscarmellose was compacted and milled on a Hosokawa Bepex Pharmapaktor APC C 80/20 (Hosokawa, Augsburg, Germany), with 80 mm diameter and 20 mm width rolls, operated at a roll speed of 10 rpm, a pressing force of 3 kN/cm and a roll gap of 1 mm. The granules were collected then screened to obtain the fraction with an adequate particle size for tableting.

### 3.6. Screening

The obtained granules from the wet- and dry-granulation processes were subjected to a screening operation to select the fraction with the desired particle size for compression. For screening, a three-sieve stack was prepared. The mesh sizes of the sieves were 1.0 mm, 0.6 mm, and 0.3 mm. All sieves were made of stainless steel (CISA S.p.A., Faenza, Italy). Screening was performed under agitation with a Heron vibrating system (Heron Aviation, Lauchringen, Germany). Any particles above the 1.0 mm sieve or below the 0.3 mm sieve were considered coarse or fine, respectively, and discarded. The fraction retained between 1.0 and 0.6 mm was selected as the compression granules. To improve the compression process, 20% (*w*/*w*) of the 0.3–0.6 mm fraction (relative to the total compression granules) was added to the compression granules. The compression granules were subsequently blended with the remaining sodium croscarmellose and glyceryl behenate, as described in the blending section, to obtain the compression blend.

### 3.7. Tablet Manufacturing

Tablets were produced on a J. Bonal Model B No. 508 press (Bonals Technologies Spain S.A., Barcelona, Spain) using concave punches (set No. 3) to yield 700 ± 35 mg, 13 mm tablets at 30 rpm (≈60 tablets/min). The compression setting was adjusted in situ for each batch to the highest feasible notched position within a narrow operating band (5.0–6.0 Bonals units; nominal target ≈ 5.5), as constrained by the press’s analog, notched lever with 0.5-unit increments. This strategy provided comparable compaction conditions across formulations under the instrument’s discrete control.

### 3.8. Characterization

#### 3.8.1. Intermediate Products

To assess suitability for compression, the flowability of each compression blend was characterized. Moreover, since the direct-compression blends were fed directly to the tablet press, their compressibility was also determined.

##### Flowability

Flowability was characterized by two parameters: the angle of repose and the flow time. Both parameters were established following the method described in the RFE [43]. As described in this method, a funnel was positioned perpendicularly at a fixed height of 20 cm above a flat surface. In all experiments, 10 g of the compression blend was placed in the funnel, with the outlet initially closed. After the blend was placed, the outlet was opened to allow the blend to flow freely onto the flat surface. The mean of two perpendicular diameters and the pile height were measured in triplicate. The angle of repose was calculated as follows:$$\tan(\alpha )=\frac{height}{0.5\;base}$$
where *α* is the angle of repose expressed in grades (°), height is the maximum pile height, measured in centimeters, and base is calculated from the corrected mean diameter in centimeters. The corrected mean diameter was determined by subtracting the diameter occupied by the material and the diameter of the stem opening of the funnel used for the determination.

The flow time is the time, in seconds, required for the blend to discharge completely after opening the funnel outlet, with no material remaining in the funnel. Flow time was recorded for each replicate.

##### Compressibility

To evaluate compressibility of direct-compression blends, *IC*, which correlates both the flux and compressibility of a substance, and *IH*; were calculated using the following equations:$$IH=\;\frac{\rho_{tapped}}{\rho_{bulk}}$$$$IC=\frac{\rho_{tapped}-\rho_{bulk}}{\rho_{bulk}}\times 100$$
where *IH* is Hausner’s ratio, which is dimensionless, and *IC* is Carr’s index as a percentage; *ρ_tapped_* is the density (g/mL) of the compression blend after being subjected to forced compaction using a PT-TD200 (Pharmatest Services Ltd., Turku, Finlandia) equipment per USP specifications; and *ρ_bulk_* is the untapped bulk density (g/mL).

#### 3.8.2. Tablets

The dimensions, height and diameter, of 20 tablets per formulation were measured with a digital caliper (Dasqua, Lodi, Italy).

##### Weighing

The developed tablets were weighed on a Mettler AM10 (Mettler-Toledo International Inc., Greifensee, Suiza) precision balance, and the mean weight and deviation were calculated. in accordance with the chapter 2.9.5 of the RFE [44]. Radiogardase^®^ mean content weight was also determined according to RFE 2.9.5 by subtracting the capsule-shell mass from the total capsule mass for each unit, prior to averaging.

##### Disintegration

Disintegration was performed following the assay described in Chapter 2.9.1 of the RFE [45] using a Pharmatest PTZ-E disintegrator (Pharmatest Services Ltd., Turku, Finlandia). Six tablets per formulation were placed individually in one tube of the disintegration basket and then subjected to the test using distilled water at 37 ± 2 °C as the medium. The time for complete disintegration of each unit (no residue in the tubes) was recorded, and mean values were calculated.

##### Hardness

Resistance to crushing (hardness) was determined in accordance with Chapter 2.9.8 of the RFE [46]. The breaking strengths of 10 tablets were measured using a Pharmatest durometer type PTB 311E (Pharmatest Services Ltd., Turku, Finlandia). Results are reported in newtons. An internal development target of NLT 39.2 N was used to ensure robustness and was trended alongside friability [47].

##### Friability

Tablet friability was determined according to Chapter 2.9.7 of the RFE [48]. Because the mean tablet weight exceeded 650 mg, 10 units were tested. The tablets were first dedusted and weighed, then rotated 100 times at 25 rpm in a Pharmatest PTF-E friability tester (Pharmatest Services Ltd., Turku, Finlandia). After the test, tablets were dedusted and reweighed, and the percentage mass loss was recorded as friability.

#### 3.8.3. Assay and Content Uniformity

The active substance content of the PB tablets was determined by quantifying total Fe and converting it to PB equivalent. Approximately 50 mg of finely powdered tablet were accurately weighed into a Teflon^®^ microwave-digestion vessel (Chemours Company, Wilmington, DE, USA) and mixed with 3 mL of HNO_3_ and 3 mL of HCl (trace-analysis grade). Samples were digested in a microwave system at 200 °C for 20 min, then diluted to 50 mL with Milli-Q^®^ water. An aliquot of 0.25 mL of this digest was further diluted to 50 mL (overall dilution 1:200) before analysis.

Iron was quantified using inductively coupled plasma-optical emission spectroscopy (ICP-OES, Varian 720, Varian, Dättwil, Switzerland) under the following operating conditions: RF power 1 kW; plasma flow 15.0 L min^−1^; auxiliary flow 1.5 L min^−1^; nebulization flow 1.0 L min^−1^; reading time 10 s.

Calibration was performed with Fe standards of 0.40, 1.00 and 2.00 mg L^−1^ prepared from a 100 mg L^−1^ commercial stock solution. Fe emission lines 238.204 nm and 259.940 nm were monitored.

The result was then expressed as % of label claim in PB equivalent, transforming %Fe into %PB with the molecular formula of PB and confirming water content by TGA [21]. In the absence of a pharmacopeial monograph for PB, and in line with ICH Q6A principles for setting specifications, the assay acceptance range was set at 95.0–105.0% of label claim [49].

Content uniformity of the 500 mg PB tablets was evaluated in accordance with the procedure described in the European Pharmacopeia (Ph. Eur.) general chapter 2.9.40 “Uniformity of dosage units” [50].

Ten tablets were selected at random, and each was individually assayed for PB content, expressed as % of the label claim.

PB content was determined by quantifying total iron (Fe) after complete sample digestion (see Assay subsection) using Inductively Coupled Plasma (ICP) spectrometry (ICP-OES).

The *AV* was calculated according to the Ph. Eur. formula:$$AV=\left|M-\bar{X}\right|+\kappa s$$
where
$\bar{X}$ = mean individual content (% of label claim)s = sample standard deviationκ = 2, 4 for *n* = 10M = reference value defined in Table 2.9.40-2 of the Ph. Eur. (98.5% when $\bar{X}$ < 98.5%; 101.5% when $\bar{X}$ > 101.5%; otherwise $\bar{X}$

According to pharmacopeial requirements, batch complies if *AV* ≤ 15.0. If *AV* > 15.0 for the initial 10 units, an additional 20 tablets are tested, and the batch is accepted if the final *AV* ≤ 15.0 and no individual content lies outside ± 25% of M.

#### 3.8.4. Dispersion (Surrogate Dissolution) Test

To evaluate the in vitro release behavior of the insoluble PB and compare it with the marketed product Radiogardase^®^, a dispersion test was performed following the general procedure of the RFE (RFE 2.9.3 “Ensayo de disolución de las formas farmacéuticas sólidas”), adapted as a surrogate dissolution assay [51].

Six PB tablets and six Radiogardase^®^ capsules were individually tested (*n* = 6) using the RFE Apparatus 1 (equivalent to USP paddle method, Apparatus 2) in simulated gastric fluid without enzymes, pH 1.2, at 37 ± 0.5 °C and 100 rpm in an Erweka DT 6 (Erweka, Langen, Germany). Sampling times were 5, 15, 30 and 60 min. At each time point, with the medium kept under constant vortex stirring to avoid sedimentation, 0.5 mL aliquots were withdrawn and immediately subjected to the same microwave-assisted acid digestion and ICP–OES assay described in Section 3.8.3. The concentration of Fe measured in the digested aliquots was converted to % of theoretical PB content. Results are expressed as mean ± SD.

### 3.9. Yield Definitions and Calculations

Process yields were defined as follows:Compression yield (%) = (mass of compressed tablets/mass of compression blend loaded) × 100.Overall tablet yield (%) = (mass of specification-conforming tablets/theoretical mass to be compressed after intermediate losses) × 100.

Route-specific step yields:Wet granulation (WG): Granulation yield (%) = (useful wet mass/initial dry blend) × 100; drying yield (%) = (dried granules/wet mass) × 100.Dry granulation (DG): Compaction–milling yield (%) = (post-milling granules/initial blend) × 100.

Screening yields (%) reported by size fraction (>1.00 mm; 0.60–1.00 mm; 0.30–0.60 mm; <0.30 mm). The target-fraction yield for tableting was the 0.60–1.00 mm fraction plus 20% (*w*/*w*) of the 0.30–0.60 mm fraction, calculated relative to the mass of the 0.60–1.00 mm fraction.

Note. Batches DC1–DC4 were exploratory and discontinued; yields were not recorded. Yield data are reported for DC5 (development and pilot lots), WG, and DG.

### 3.10. SeDeM Methodology Study

In addition to the studies performed on the intermediate products and developed tablets, a SeDeM study of the DC5 formulation, which showed superior performance during development, was performed. The objective was to predict and confirm the suitability of the selected formulation for scale-up to industrial manufacturing. The SeDeM methodology is a formulation tool that allows assessment of direct-compression formulations for future in industrial tablet manufacturing. To achieve this, SeDeM uses five parameters known as incident factors, which are obtained mathematically from twelve parameters measured or calculated in the API-excipient mixture [52].

The experimental parameters of the SeDeM methodology are as follows:Bulk density (*Da*): *Da* was determined as described in the compressibility subsection.Tapped density (*Dc*): *Dc* was also determined in the same way as in the compressibility subsection.Interparticle porosity (*Ie*): dimensionless parameter, calculated from Da and Dc as: $$Ie=\frac{Dc-Da}{Dc\times Da}$$Carr index (*IC*): calculated as described in the compressibility subsection.Cohesion index (*Icd*): the measure was performed as specified in the tablets characterization subsection.Hausner ratio (*IH*): *IH* was calculated using the equation in the compressibility subsection.Angle of repose (*α*): the angle of repose was measured as indicated in the flowability subsection, and its determination followed the methodology specified in the RFE [43].Flow time (t’’): the flow time was measured as described in the flowability subsection.Loss on drying (%HR): determined per Ph. Eur. was followed [53]. The sample was dried at 105 ± 2 °C in a Selecta Furnace model 207 (J.P. Selecta, S.A.U., Abrera, Spain) until a stable weight was reached.Hygroscopicity (%H): increase in sample weight after exposure to 76 ± 2% RH at 22 ± 2 °C for 24 h.Particles < 50 µm (%Pf): 50 g of product were weighed and sieved at 50 µm for 10 min under constant agitation; the percentage (*w*/*w*) passing was calculated.Homogeneity index (Iθ): 50 g of the API-excipient blend was sieved through a stack with decreasing mesh sizes of 355 µm, 212 µm, 100 µm and 50 µm for 10 min at constant vibration. Once screened, each fraction was weighed, and the homogeneity index was calculated using the following equation:
$$I\theta =\frac{Fm}{100+\left(dm-dm-1\right)\times Fm-1+\left(dm+1-dm\right)\times Fm+1+\left(dm-dm-n\right)\times Fm-n+\left(dm+n-dm\right)\times Fm+n}$$
where Fm is the% (*w*/*w*) of particles in the majority range; Fm − 1 is the% (*w*/*w*) of particles in the rank immediately below the majority range; Fm + 1 is the% (*w*/*w*) of particles in the range immediately above the majority range; n is the position number of the studied fraction with respect to the majority fraction; dm is the mean diameter of the particles in the majority fraction; dm − 1 is the mean diameter of the particles in the fraction immediately below the majority fraction; and dm + 1 is the mean diameter of the particles in the fraction immediately above the majority fraction.

After establishing the parameter values, they were mathematically transformed into r parameters using conversion factors applied to the original values to obtain r parameters with values between 0 and 10 (Table 11).

Once calculated, the r parameters are graphically represented in an SeDeM diagram (Figure 8).

To determine the suitability of the compression blend, incidence factors are used. Each of the r parameters is used to calculate one of the five incidence factors [54,55].

The dimensional incidence factor (F_dimensional_) is the stacking capacity of the powder and its effects on the tablet dimensions; this is defined as follows: F_dimensional_ = mean (Da; Dc)The compressibility incidence factor (F_compressibility_) is the capacity of the blend to be compressed and retain its form; this is expressed as follows: F_compressibility_ = mean (Ie; IC; Icd)The flowability incidence factor (F_flowability_) is the flow capacity of the compression blend and is defined as follows: F_flowability_ = mean (IH; *α*; t’’)The lubrication/stability incidence factor (F_lubrication/stability_) relates to the residual humidity and the blend capacity to capture it with its flow and compactness and is expressed as follows: F_lubrication/stability_ = mean (%HR; %H)The lubrication/dosage incidence factor (F_lubrication/dosage_) relates the particle size distribution with the capacity of the blend to flow and the proper filling of compression mattresses; this is defined as follows: F_lubrication/dosage_ = mean (%Pf; Iθ)

From the incidence factors, three indices are calculated to determine if the blend is suitable for a direct compression process: the parametric index, the parametric profile index and the good compression index [56,57].

6.The Parametric Index (IP) is calculated using the following equation: $$IP=\frac{nºP\geq 5}{nºPt}$$where *nº P* ≥ 5 is the number of parameters which value is ≥ 5; and *nº Pt* is the total number of examined parameters. The minimal value for a good powder mixture is an *IP* ≥ 0.5.7.The parametric profile index (IPP) is calculated using the following equation:$$IPP=\frac{\sum_{i=1}^{n}ri}{n}$$
where $\sum_{i=1}^{n}ri$ is the addition of the radium value of all examined parameters; and n is the total number of examined parameters. The minimum value desirable for appropriate powder characteristics is *IPP* ≥ 5.8.The Good Compression Index (*IGC*) is calculated using the following equation:IGC=IPP×f
where *IPP* is the parametric profile index; and F is the reliability factor *(f* = 0.952). The minimum value expected for good direct compression is *IGC* ≥ 5.

### 3.11. Cs Binding Capacity of the Tablets

To determine the Cs-binding capacity of the PB tablets, an in vitro procedure was used. For the test, one PB tablet was added to 100 mL of phosphate buffer (40 mM, pH 7.5) containing 100 ppm Cs; the suspension was agitated for 24 h at 37 °C. After agitation, the suspension was filtered, and the unbound Cs was measured using inductively couple plasma-optical emission spectroscopy (ICP-OES, Varian 720 instrument, Varian Inc., Palo Alto, CA, EE. UU.). As a reference, the same procedure was carried out with one Radiogardase^®^ capsule, the marketed medicinal product containing PB as the active ingredient. The ICP-OES operating conditions were as follows:Potency: 1 kW.Plasma flow: 15.0 L/min.Auxiliary flow: 1.5 L/min.Nebulization flow: 1.0 L/min.Reading time: 10 s.Wavelength: 697.3 nm.

### 3.12. Stability Study

To assess the quality and efficacy of PB tablets over time, a long-term stability study was designed and performed. ICH guide instructions were followed for the design, implementation and execution of the stability studies [17]. According to the ICH climatic zone classification, the stability study was performed in Climatic Zone II conditions (25 ± 2 °C/60 ± 5% RH) for 5 years from study initiation. An accelerated study was also performed for 6 months at 40 ± 2 °C and 75 ± 5% RH, as defined in the ICH guidelines. For both conditions, tablets from three batches were stored for the study duration in climatic chambers (one per condition), meeting the specified temperature and humidity requirements. The tablets stored for the study were packaged in PVC/PVDC-aluminum blisters.

Following ICH guideline recommendations, the number of tablets needed for stability tests was removed from storage at 0, 3, 6, 9, 12 and 18 months for the long-term conditions and 0, 3 and 6 months for the accelerated conditions. The quality and efficacy characteristics of the tablets tested at the defined times were the description, mean weight, hardness, humidity and decorporation capacity. All tests were carried out as specified in the RFE, except for the% humidity and decorporation capacity [46,56]. The tests were performed as described below for each batch:Description: samples were visually evaluated and compared to the specifications established during the batch manufacturing. The descriptions included the color, integrity, existence of defects, and shape.Mean weight: Twenty tablets were weighed on a Mettler AM10 (Mettler, Spain) precision balance, and the mean weight was calculated. The maximum weight deviation accepted was 5% from the tablet nominal weight of 700.0 mg. Moreover, no tablet could be above 10% deviation, and no more than 2 of the 20 tablets could deviate by 5–10%.% weight loss by thermogravimetric analysis (TGA): Due to the importance of the water content in the PB mechanism of action, the amount of water in PB during the stability test and the type of water that it possessed needed to be determined. The water content was measured with a GA55 TGA Analyzer (TA Instruments, New Castle, Estados Unidos, DE, USA) at a heating rate of 20 °C/min on a ramp from 25 °C to 360 °C, with N_2_ applied at a constant flow rate of 40 mL/min.Cs binding capacity: The decorporation capacity of the tablets tested in the stability study was determined using the same method in the efficacy subsection. The acceptance criterion was no less than the initial Cs decorporation capacity minus 10%.

Microbial contamination was evaluated in nonsterile products according to the procedures outlined in the RFE, Chapters 2.6.12 (“Control of microbial contamination in nonmandatory sterile products—Total Aerobic Microorganism Count”) and 2.6.13 (“Control of microbial contamination in nonmandatory sterile products—Specific Microorganism Assays”) [57,58]. The following assays were performed at time 0 and after 12 months of storage:Total aerobic microorganism count (TAMC): Sample preparation: A 1 g sample of the product was aseptically weighed and transferred into a sterile container containing 9 mL of sterile diluent (buffered peptone water). The sample was homogenized using a stomacher to ensure complete dispersion of the microorganisms. Serial dilutions and plating: Serial decimal dilutions were prepared in sterile diluent. A 1 mL aliquot from the appropriate dilution was plated onto plate count agar using the pour plate method. Plates were incubated at 30–35 °C for 72 h. Enumeration: After incubation, the colonies were counted, and results are expressed as colony-forming units per gram (CFU/g). According to the RFE, the test was considered satisfactory if the CFU/g did not exceed 10^3^.Total yeast and mold count (TYMC): Sample preparation and dilution: The same initial sample preparation and serial dilution procedure used for the TAMC was applied. Plating: An aliquot from a suitable dilution was plated onto Sabouraud dextrose agar supplemented with chloramphenicol to inhibit bacterial growth. Incubation: The plates were incubated at 25 °C for 5 days to allow for the development of yeast and mold colonies. Enumeration: The colonies were counted, and the results are expressed as CFU/g. The test was considered satisfactory if the count did not exceed 10^2^ CFU/g.The assays for specific microorganisms were conducted for both *E. coli* and *C. albicans* as described in Chapter 2.6.13 [58].Detection of *Escherichia coli*: Plating: An aliquot (typically 1 mL) from the appropriate dilution was inoculated onto MacConkey agar plates. Incubation: The plates were incubated at 37 °C for 24 h. Interpretation: The plates were examined for the presence of typical *E. coli* colonies. The test was considered satisfactory if no *E. coli* colonies were observed in any of samples.Detection of *Candida albicans*: Plating: Similarly, an aliquot was inoculated onto Sabouraud agar supplemented with chloramphenicol. Incubation: The plates were incubated at 25–30 °C for 48–72 h. Interpretation: The presence of *C. albicans* was noted. For this study, the method validation focused on ensuring that the media supported the growth of the reference strain, as described below.

Validation of media nutritional and selective properties: Prior to testing, the nutritional and selective properties of the media were verified. Known reference strains of *Escherichia coli* and *Candida albicans* were inoculated onto MacConkey agar and Sabouraud agar with chloramphenicol, respectively, both in the presence and absence of the test product. This validation step confirmed that the culture media not only provided sufficient nutritional support for the growth of the target microorganisms but also retained their selective properties, thereby ensuring that only the microorganisms of interest were detected under the assay conditions.

At the time of writing, the study is ongoing; claims in the manuscript are limited to completed time points 18 months).

## 4. Conclusions

The design and development of tablets for the oral administration of PB as a decorporant for radioactive and toxic species provides a new capacity for the treatment of casualties generated in nuclear and radiological incidents and accidents. In this study, we developed three tablet formulations containing PB; to our knowledge, PB tablets have not previously been marketed or reported. Each formulation (DC5, WG and DG) is designed for one of the main and more advanced tableting processes: direct compression or compression of granules obtained by wet granulation or dry granulation. This wide range of formulations facilitates potential technology transfers for tablet manufacturing by any of the described methods.

The formulation of PB tablets by direct compression is distinguished from the other formulations tested because of its superior technological advantages of ease, speed and high production capacity; thus, this process is recommended as the optimal choice for its industrial production and accomplishes the exigencies of the Spain Ministry of Defense stipulated in the AZPINDECS project. The safety, efficacy, and quality characteristics of the PB tablets over 18 months were demonstrated in a stability study performed according to ICH guidelines. Stability studies at 24, 36, 48, and 60 months are ongoing. Furthermore, PB tablet formulation effectively captures and retains the target chemical species.

## Figures and Tables

**Figure 1 pharmaceuticals-18-01568-f001:**
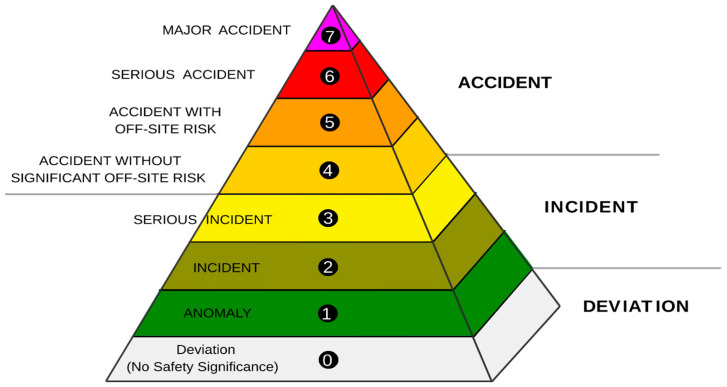
INES for classifying nuclear events [7].

**Figure 2 pharmaceuticals-18-01568-f002:**
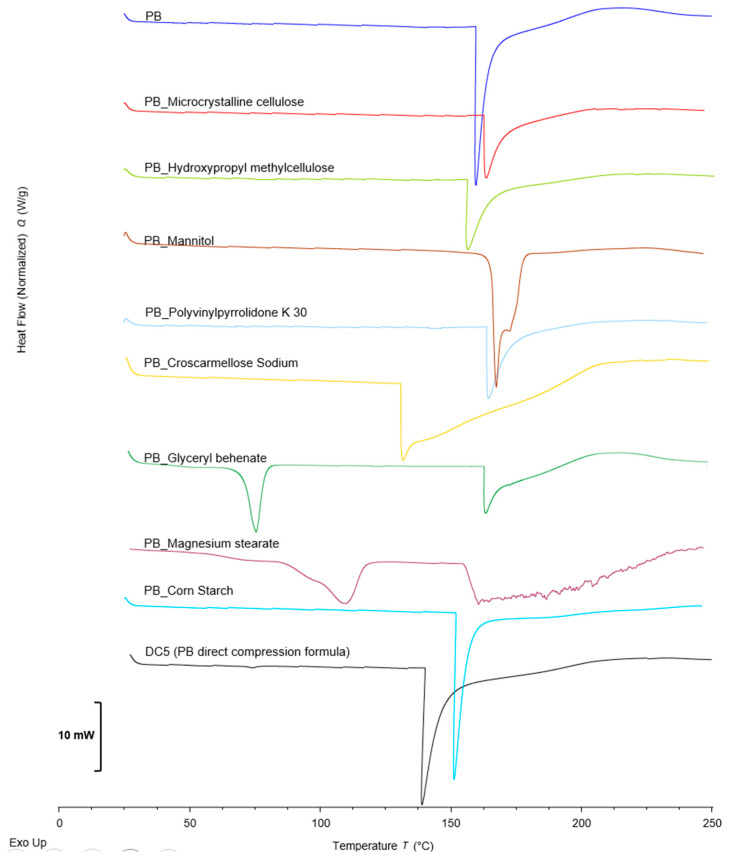
DSC-thermograms of the PB raw material, the physical binary mixtures of PB and each of the excipients proposed (1:1) and the direct compression blending DC5.

**Figure 3 pharmaceuticals-18-01568-f003:**
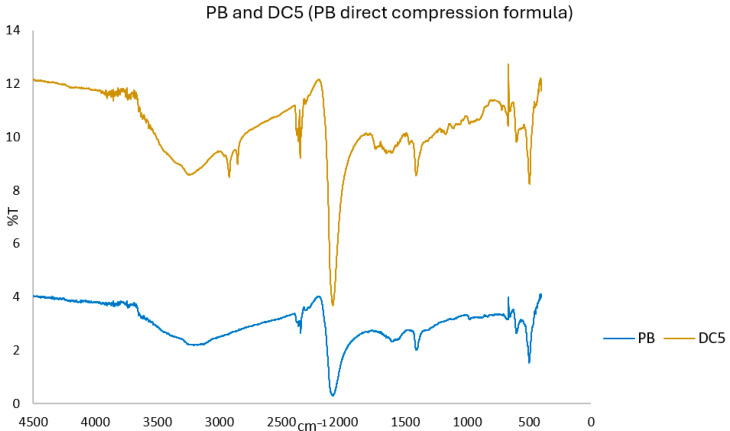
Fourier-transform infrared (FTIR) spectra of PB and the final direct-compression formulation DC5.

**Figure 4 pharmaceuticals-18-01568-f004:**
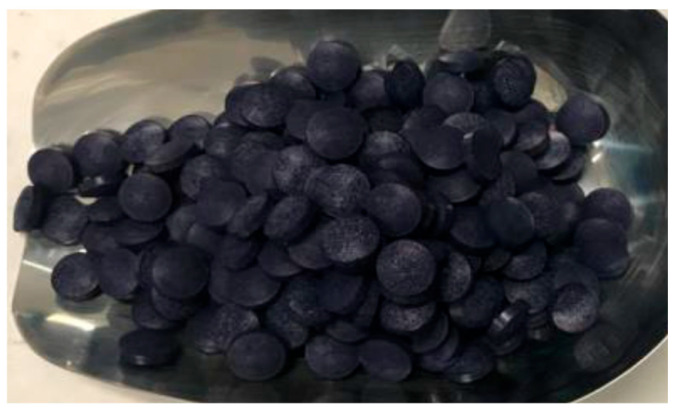
Final selected tablets produced via direct compression following the DC5 formula.

**Figure 5 pharmaceuticals-18-01568-f005:**
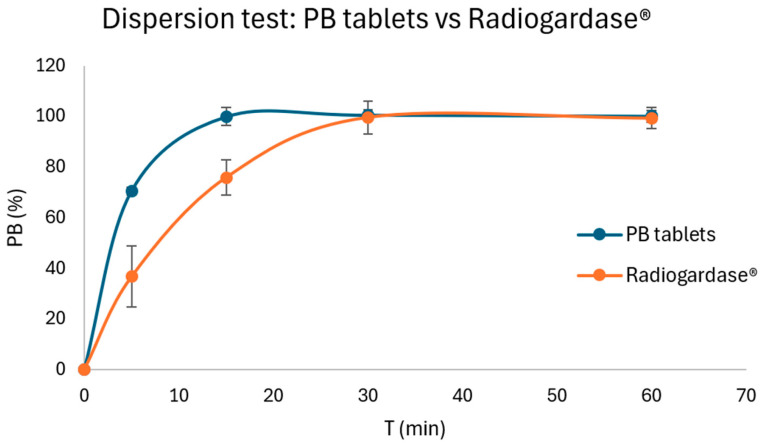
Mean percentage of PB dispersed (± SD, *n* = 6) over time in simulated gastric fluid without enzymes (pH 1.2) at 37 ± 0.5 °C, using the Real Farmacopea Española (RFE) Apparatus 1 (equivalent to USP paddle method) at 100 rpm. Two samples are presented: PB tablets (blue) and Radiogardase^®^ (orange).

**Figure 6 pharmaceuticals-18-01568-f006:**
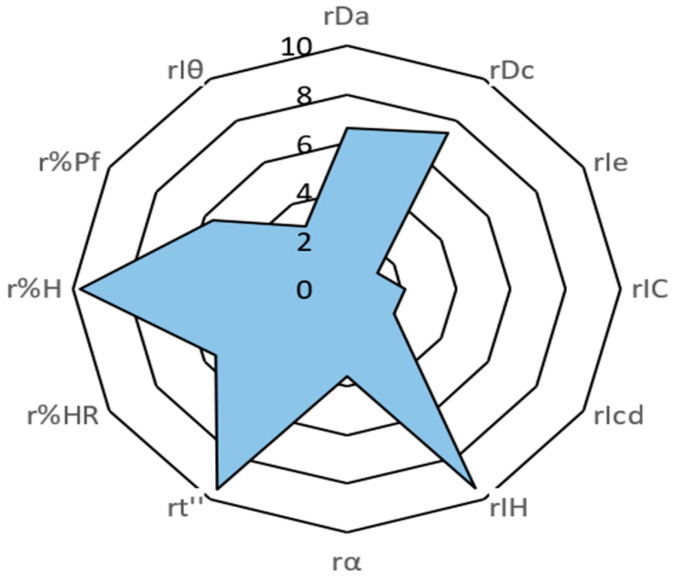
SeDeM diagram of the DC5 intermediate product.

**Figure 7 pharmaceuticals-18-01568-f007:**
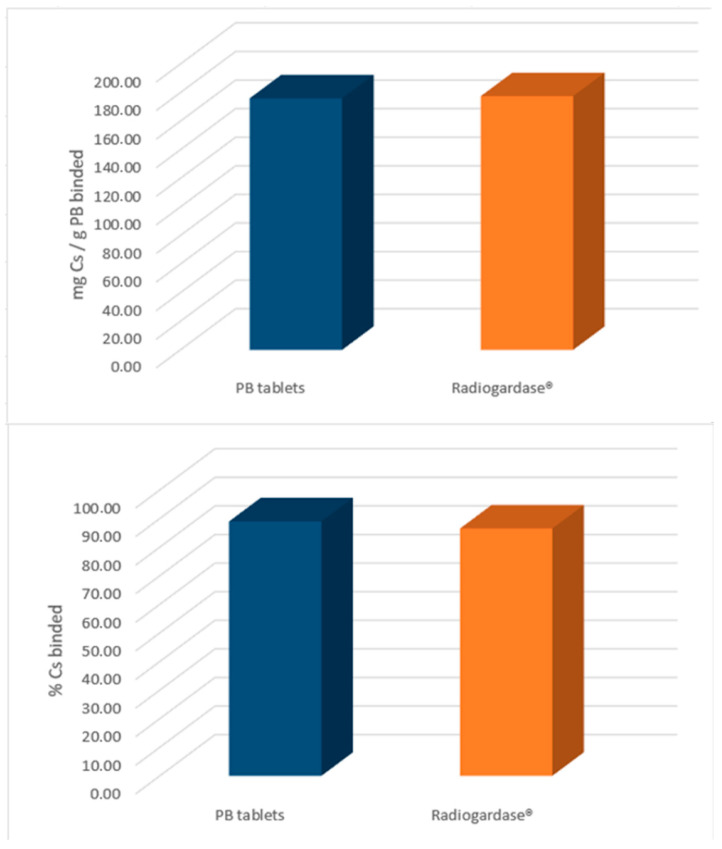
Cs binding capacity of the PB tablets (blue) and 500 mg Radiogardase (orange) hard capsules. On the **top**: Cs bound in mg per gram of PB. On the **bottom**: %Cs bound per tablet or hard capsule.

**Figure 8 pharmaceuticals-18-01568-f008:**
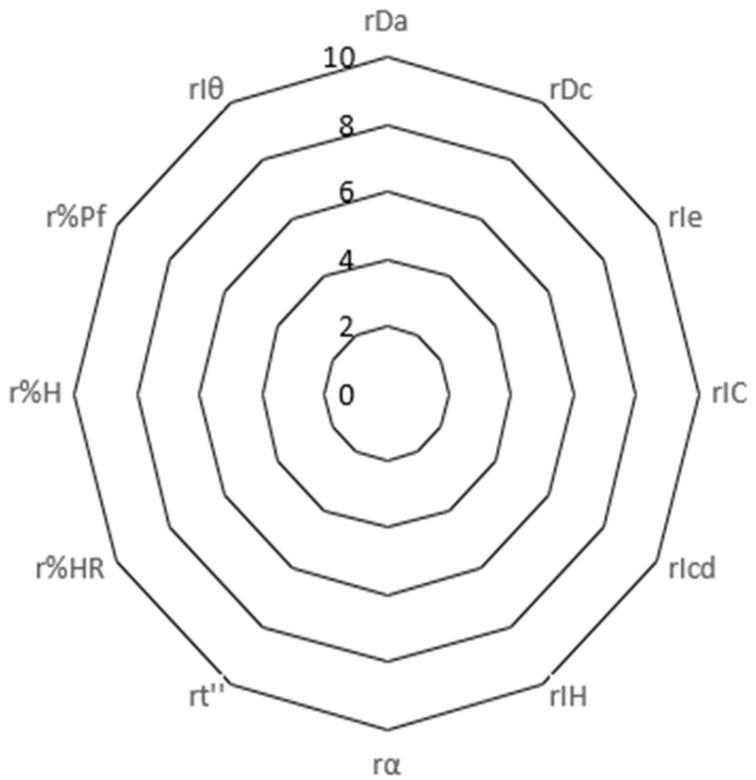
SeDeM standard diagram.

**Table 1 pharmaceuticals-18-01568-t001:** Formulations for the tablets manufactured by direct compression.

Component	DC1 (%)	DC2 (%)	DC3 (%)	DC4 (%)	DC5 (%)
Prussian blue	71.43	71.43	71.43	71.43	71.43
Microcrystalline cellulose	11.32	11.32	11.32	11.79	23.57
Mannitol	11.32	11.32	11.32	11.78	0.00
Povidone K30	3.00	3.00	3.00	3.00	3.00
Sodium croscarmellose	1.43	1.43	1.43	0.00	0.00
Glyceryl behenate	0.50	0.00	0.25	1.00	1.00
Magnesium stearate	0.00	0.50	0.25	0.00	0.00
Hydroxypropyl methylcellulose	1.00	1.00	1.00	1.00	1.00
Total	100.00	100.00	100.00	100.00	100.00

**Table 2 pharmaceuticals-18-01568-t002:** Formulation for the manufacturing of the tablets produced using wet granulation.

Component	WG (%)
Prussian blue	71.43
Microcrystalline cellulose	21.07
Povidone K30	5.50
Corn starch	1.0
Glyceryl behenate	1.0
Total	100.00

**Table 3 pharmaceuticals-18-01568-t003:** Formulation for the manufacturing of the tablets produced using a dry granulation process.

Component	DG (%)
Prussian blue	71.43
Microcrystalline cellulose	23.57
Povidone K30	3.00
Sodium croscarmellose	1.0
Glyceryl behenate	1.0
Total	100.00

**Table 4 pharmaceuticals-18-01568-t004:** Results of the intermediate product analysis for all formulations.

Formula	Angle of Repose *α* (°)	Flow Time (s)	Hausner Ratio (IH)	Carr Index IC (%)
DC1	32.61	1.2	1.13	11.76
DC2	34.22	1.8	1.16	13.76
DC3	33.59	1.5	1.15	12.66
DC4	32.30	1.0	1.13	11.64
DC5	32.11	1.0	1.11	10.81
WG	32.05	1.0	1.12	10.42
DG	34.40	1.0	1.14	12.37

**Table 5 pharmaceuticals-18-01568-t005:** Flow and compressibility properties [37].

Flow and/or Compressibility Properties	Angle of Repose *α* (°)	Hausner Ratio (IH)	Carr Index IC (%)
Excellent	25–30	1.00–1.11	25–30
Good	31–35	1.12–1.18	31–35
Fair	36–40	1.19–1.25	36–40
Passable	41–45	1.26–1.34	41–45
Poor	46–55	1.35–1.45	46–55
Very poor	56–65	1.46–1.59	56–65
Extremely poor	>66	>1.60	>66

**Table 6 pharmaceuticals-18-01568-t006:** Results obtained after tablet analysis.

Formula	Description	Mean Weight (mg)	Diameter (cm)	Height (cm)	Disintegration Time (min)	Hardness (N)	Friability (%)
DC1	Blue tablets, with some visible defects and small breaks, round, biconvex.	696.0 ± 3.4	1.3	0.39 ± 0.02	<1	31.8	Test failed, cracked tablets
DC2	Blue tablets, with some visible defects and small breaks, round, biconvex.	695.9 ± 4.0	1.3	0.45 ± 0.03	<1	26.4	Test failed, broken tablets
DC3	Blue tablets, with some visible defects and small breaks, round, biconvex.	698.0 ± 3.7	1.3	0.40 ± 0.04	<1	29.8	Test failed, broken tablets
DC4	Blue tablets, with some visible defects and small breaks, round, biconvex.	698.4 ± 3.1	1.3	0.38 ± 0.02	<1	36.7	1.05 (failed test due to result > 1%)
DC5	Blue tablets, without breaks or defects, round, biconvex.	699.8 ± 3.3	1.3	0.38 ± 0.01	<1	41.3	0.82
WG	Blue tablets, without breaks or defects, round, biconvex.	700.2 ± 3.4	1.3	0.35 ± 0.01	<1	46.6	0.77
DG	Blue tablets, without breaks or defects, round, biconvex.	696.0 ± 3.0	1.3	0.43 ± 0.03	<1	39.0	0.96
Radiogardase^®^ *	Opaque, very dark blue hard capsules, cap and body of the same color. Cylindrical with hemispherical ends, no cracks, breaks or other visible defects. Heyl and PB imprints visible.	496.5 ± 9.7 (length)	2.1	0.73 ± 0.01	9.5 ± 1.8	N/A	N/A

* Commercial Radiogardase^®^ (hard capsules) is included to facilitate comparison with the PB tablet formulations. Pharmacopeial hardness (resistance to crushing) and friability tests are defined for tablets, not for hard capsules; therefore, no hardness or friability results are reported for Radiogardase (N/A).

**Table 7 pharmaceuticals-18-01568-t007:** Experimental results and calculated parameters, incidence factors and indices of the DC5 formulation intermediate product.

Experimental Results											
Da (g/mL)	Dc (g/mL)	Ie	IC (%)	Icd (N)	IH	*α* (°)	t’’ (s)	%HR	%H	%Pf	Iθ
0.66	0.74	0.16	10.81	40.20	1.11	32.11	1.00	4.50	0.48	21.86	0.01
**Parameters (R)**											
rDa	rDc	rIe	rIC	rIcd	rIH	r*α*	rt’’	r%HR	r%H	r%Pf	rIθ
6.60	7.40	1.33	2.16	2.01	9.45	3.58	9.50	5.50	9.76	5.63	2.99
**Incidence factors**											
Dimensional		Compressibility			Flowability			Lubrication/ stability		Lubrication/ dosage	
7.00		1.84			7.51			7.63		4.31	
**Acceptance indices**											
Parametric Index (IP)				Parametric Profile Index (IPP)				Good Compression Index (IGC)			
0.58				5.49				5.23			

**Table 8 pharmaceuticals-18-01568-t008:** Stability studies for the PB tablets under accelerated conditions.

PB Tablets Accelerated Conditions Stability Results								
Time of Analysis	Description: “Blue Tablets, Without Breaks or Defects, Round, Biconvex”	Mean Weight (mg): “700.0 mg ± 5% (665–735 mg)”	Hardness (N) “No Less than 39.2”	Friability (%) “No More than 1.00%”	Disintegration (min) “No More than 15 min”	% Weight Loss: “No Less than 25.00%”	Cs Binding Capacity: “No Less than 79%”	Assay (% Label Claim) “95–105%”
Initial (t = 0)	Comply	701.9 ± 8.3	41.7 ± 0.2	0.80	<1 min	35.72 ± 4.62	89 ± 1	-
3 months	Comply	702.3 ± 11.8	41.0 ± 0.5	0.82	<1 min	42.23 ± 1.77	89 ± 1	-
6 months (end of the accelerated conditions study)	Comply	700.7 ± 13.1	40.9 ± 0.4	0.85	<1 min	35.98 ± 4.24	88 ± 1	-

**Table 9 pharmaceuticals-18-01568-t009:** Results from the stability studies in climatic zone II conditions for the PB tablets at 18 months.

PB Tablets Climatic Zone II Conditions Stability Results								
Time of Analysis	Description: “Blue Tablets, Without Breaks or Defects, Round, Biconvex”	Mean Weight (mg): “700.0 mg ± 5% (665–735 mg)”	Hardness (N) “No Less than 39.2”	Friability (%) “No More than 1.00%”	Disintegration (min) “No More than 15 min”	% Weight Loss: “No Less than 25.00%”	Cs Binding Capacity: “No Less than 79%”	Assay (% Label Claim) “95–105%” *
Initial (t = 0)	Comply	701.9 ± 8.3	41.7 ± 0.2	0.80	<1 min	35.72 ± 4.62	89 ± 1	-
3 months	Comply	699.5 ± 16.1	41.5 ± 0.4	0.80	<1 min	33.47 ± 5.53	89 ± 2	-
6 months	Comply	699.6 ± 12.7	41.1 ± 0.2	0.82	<1 min	32.20 ± 0.58	85 ± 4	-
9 months	Comply	700.5 ± 9.4	40.7 ± 0.3	0.83	<1 min	34.38 ± 3.50	86 ± 3	-
12 months	Comply	702.3 ± 14.6	40.2 ± 0.5	0.84	<1 min	39.58 ± 3.88	88 ± 2	-
18 months	Comply	699.4 ± 10.2	39.8 ± 0.3	0.86	1.17 min ± 0.12	32.85 ± 2.52	87 ± 4	97.3 ± 1.2

* In response to the reviewers’ recommendation, a validated assay for the active substance content of the PB tablets was developed and included in the product specification. Assay testing has been applied to all available stability samples; at present, data are available up to 18 months. Because the ongoing ICH long-term stability study is designed for a total duration of 60 months, additional assay data will be generated.

**Table 10 pharmaceuticals-18-01568-t010:** Microbiological stability studies for the PB tablets under accelerated conditions.

PB Tablets Climatic Zone II Conditions Stability Results				
Time of Analysis	Climatic Zone Conditions	Total Aerobic Microorganisms Count (TAMC) NMT 10^3^ CFU/g	Total Yeast/Mold Count (TYMC) NMT 10^2^ CFU/g	*Escherichia coli* in 1 g Negative
Initial (t = 0)	25 ± 2 °C 60 ± 5% RH	Comply	Comply	Comply
	40 ± 2 °C 75 ± 5% RH	Comply	Comply	Comply
12 months	25 ± 2 °C 60 ± 5% RH	Comply	Comply	Comply
	40 ± 2 °C 75 ± 5% RH	Comply	Comply	Comply

**Table 11 pharmaceuticals-18-01568-t011:** Conversion factors for calculating the r parameters.

Experimental Parameters	Conversion Factor	r Parameter
Bulk density (Da)	10 × Da	rDa
Tapped density (Dc)	10 × Dc	rDc
Interparticle porosity (Ie)	(10 × Ie)/1.2	rIe
Carr index (IC)	IC/5	rIC
Cohesion index (Icd)	Icd/20	rIcd
Hausner Ratio (IH)	5 × (3 − IH)	rIH
Angle of repose (*α*)	10 − (*α*/5)	r*α*
Flow time (t’’)	10 − (t’’/2)	rt’’
Loss on drying (%HR)	10 − %HR	r%HR
Hygroscopicity (%H)	10 − (%H/2)	r%H
Particles above < 50 µm (%Pf)	10 − (%Pf/5)	r%Pf
Homogeneity index (Iθ)	500 × Iθ	rIθ

## Data Availability

The original contributions presented in this study are included in the article. Further inquiries can be directed to the corresponding author.

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
