# Peer review of "Innovative Galenic Formulation of Prussian Blue Tablets: Advancing Pharmaceutical Applications"

_pharmaceuticals, 2025, doi:10.3390/ph18101568_

Round 1
Reviewer 1 Report
Comments and Suggestions for Authors
Dear Author,
Your manuscript entitled “Innovative Galenic Formulation of Prussian Blue Tablets: Advancing Pharmaceutical Applications” has been comprehensively evaluated. The manuscript presents a relevant and timely contribution on the development of Prussian Blue tablets as an alternative to Radiogardase®. This study is well designed and provides valuable data on excipient selection, manufacturability, and stability. However, several important issues need to be addressed before the work can be considered further:
- The manuscript lacks content uniformity data for the 500 mg Prussian Blue tablets. While the dose is inferred from formulation ratios and weight uniformity, pharmacopoeial standards (Ph. Eur., USP <905>) require assay or uniformity testing to confirm consistent API content per tablet; this should be included or justified.
- Stability results up to 18 months are encouraging; however, projections for 24–60 months remain speculative. Any statements regarding long-term compliance should be moderated or explicitly presented as ongoing studies.
- Radiogardase® is the only approved PB product; however, the introduction could better highlight existing literature on alternative formulations or past attempts at tabletization. This would contextualize the novelty of the present work.
- Hardness, friability, and tensile strength are not presented. These parameters are essential in physical pharmacy to ensure robustness during handling, packaging, and transport.
- Disintegration times are reported, but no dissolution profiles or surrogate release studies are provided. Even if PB is insoluble, dissolution/dispersion testing is critical to confirm reproducible performance and comparability with Radiogardase®.
- While DSC provides useful thermal data, it does not fully capture possible chemical interactions between Prussian Blue and excipients. Among available spectroscopic techniques, FTIR represents the most straightforward and appropriate choice, as it directly detects functional group changes and potential interaction sites.
- The manuscript includes a useful abbreviations list, but its use in the text is inconsistent. Some terms (e.g., WG, DG, DC1–DC5) appear before being defined, while others (e.g., IH, IC, IGC) are listed but not applied consistently. The list should be alphabetized, abbreviations expanded at first mention, and only essential terms retained to ensure clarity.
Best regards.
Author Response
Reviewer 1
Dear Author,
Your manuscript entitled “Innovative Galenic Formulation of Prussian Blue Tablets: Advancing Pharmaceutical Applications” has been comprehensively evaluated. The manuscript presents a relevant and timely contribution on the development of Prussian Blue tablets as an alternative to Radiogardase®. This study is well designed and provides valuable data on excipient selection, manufacturability, and stability. However, several important issues need to be addressed before the work can be considered further:
- The manuscript lacks content uniformity data for the 500 mg Prussian Blue tablets. While the dose is inferred from formulation ratios and weight uniformity, pharmacopoeial standards (Ph. Eur., USP <905>) require assay or uniformity testing to confirm consistent API content per tablet; this should be included or justified.
We sincerely thank the reviewer for this valuable and constructive comment. In response, we have carefully revised the manuscript to address the issue by:
- Implemented and described an assay of PB content based on ICP-OES quantification of total iron with conversion to PB equivalent (see Section 3.8.3). In accordance with ICH Q6A, and given the absence of a PB monograph, we set an assay acceptance range of 95.0–105.0 % of the label claim.
- Included content-uniformity testing per Ph. Eur. 2.9.40 (ten individual tablets; AV calculation), reported in Section 2.5 with the methodological details in Section 3.8.3 (uniformity content tested on the fresh batcj.
- Reported the results: at release (fresh batch), assay = 101.3 % (SD = 0.65 %, n = 10) and at 18 months (Zone II), assay = 97.3 %, also within the predefined 95.0–105.0 % range.
- Aligned the finished-product specification (Supplementary Table S2) with these tests and limits and referenced it in the Results/Stability sections. Stability results up to 18 months are encouraging; however, projections for 24–60 months remain speculative. Any statements regarding long-term compliance should be moderated or explicitly presented as ongoing studies.
We thank the reviewer for this insightful remark. We have restricted firm claims to the completed time points (0–18 months in Zone II; 0–6 months accelerated) and now explicitly state that the stability study is ongoing. References to 24–60 months are prospective and are presented as ongoing assessments
Corresponding edits have been implemented in the Abstract, Results (Stability subsection 2.8), Materials & Methods (Stability Study, subsection 3.11) and Conclusions.
- Radiogardase® is the only approved PB product; however, the introduction could better highlight existing literature on alternative formulations or past attempts at tabletization. This would contextualize the novelty of the present work.
We thank the reviewer for this constructive suggestion.
We have expanded the Introduction to include a concise overview of the main alternative Prussian blue (PB) formulations described in the literature, such as PB-cellulose aerogels, magnetic hydrogel beads and nanostructured PB materials, as well as an oral PB formulation with pH-modifying excipients.
These reports illustrate the active development of PB delivery systems but also show that no peer-reviewed studies have described PB tablets prepared by direct compression, which underlines the novelty of our work.
The new paragraph and supporting references have been inserted in page 3, between paragraphs 2 and 3.
- Hardness, friability, and tensile strength are not presented. These parameters are essential in physical pharmacy to ensure robustness during handling, packaging, and transport.
We sincerely thank the reviewer for drawing attention to the importance of tensile strength as a mechanical performance indicator. In the present work we followed the testing requirements of the Real Farmacopea Española and the European Pharmacopoeia, where hardness and friability are defined tests, but tensile strength is not currently included; for this reason, tensile strength was not reported in the original manuscript.
That said, we fully appreciate the value of this parameter for a more complete mechanical characterization. We have now added hardness (N) and friability (%) results to Table 6 (Tablet analysis) and to Table 9 (Stability), and we have described the corresponding procedures in Section 3.8.2 (Methods). We are studying the diametral compression method to calculate tensile strength (σt) so it can be incorporated in future developments and publications. Section 3.8.2 (Methods): added detailed procedures for hardness and friability.
Table 6 (Tablet analysis): added columns for Hardness (N) and Friability (%).
Table 9 (Stability, Zone II): added Hardness (N) and Friability (%).
- Disintegration times are reported, but no dissolution profiles or surrogate release studies are provided. Even if PB is insoluble, dissolution/dispersion testing is critical to confirm reproducible performance and comparability with Radiogardase®.
We thank the reviewer for this insightful comment. The abbreviations have been revised for consistency, with definitions provided at first mention, the list alphabetized, and only essential terms retained. These changes are reflected in Section 3.8.4. We sincerely thank the reviewer for highlighting the importance of demonstrating the release behaviour of an insoluble active such as Prussian blue. In response, we performed a dispersion test as a surrogate dissolution assay (see new Section 3.8.4 and Results Section 2.5.3, Figure 5). The assay follows the Real Farmacopea Española (RFE) 2.9.3 dissolution test for solid oral dosage forms, using the Apparatus 1 described in the RFE (equivalent to USP Apparatus 2 - paddle method) and simulated gastric fluid without enzymes (pH 1.2). Prussian blue content in the withdrawn samples was quantified by the newly validated assay (ICP–OES after microwave digestion). This additional study confirms that the PB tablets disperse rapidly and completely, with profiles fully comparable, or even superior, to the marketed Radiogardase®. While DSC provides useful thermal data, it does not fully capture possible chemical interactions between Prussian Blue and excipients. Among available spectroscopic techniques, FTIR represents the most straightforward and appropriate choice, as it directly detects functional group changes and potential interaction sites.
We thank the reviewer for this constructive suggestion. To complement the DSC study, we have performed Fourier-transform infrared (FTIR) spectroscopy on Prussian blue (PB), the individual excipients of the DC5 formulation, their binary mixtures with PB, and the final DC5 tablet blend. The FTIR results confirm the DSC findings and provide additional evidence of the absence of chemical interactions between PB and the selected excipients.
Changes in the manuscript
- Section 2.3 has been retitled “Formula Components Compatibility Study Using Differential Scanning Calorimetry (DSC) and Fourier Transform Infrared Spectroscopy (FTIR)” and a new paragraph has been added describing the FTIR results.
- Figure 3 (main text) shows the FTIR spectra of PB and the DC5 formulation.
- Supplementary Figure S2 presents the spectra of PB with each excipient and their corresponding binary mixtures.
Section 3.3 (Methods) now includes the FTIR experimental procedure.
- The manuscript includes a useful abbreviations list, but its use in the text is inconsistent. Some terms (e.g., WG, DG, DC1–DC5) appear before being defined, while others (e.g., IH, IC, IGC) are listed but not applied consistently. The list should be alphabetized, abbreviations expanded at first mention, and only essential terms retained to ensure clarity.
This has been done in accordance with the reviewer's request.
Reviewer 2
The article addresses a definite need to have publicly available description of manufacturing process for Prussian blue tablets. However, it needs some improvement in order to deliver the content in its entirety.
Firstly, the organization of the articles is slightly confusing (methods after the results) and a great deal of attention is given to some details (e.g. DSC and SeDeM), while other parts are lacking. The article needs to be reorganized and shortened to be clearer.
In tablet manufacturing, the specification of tablets is the single most useful and critical document that defines and ensures repeatable quality and clinical efficacy of the tablets. Specification for your tablets should be provided and relied upon in the article (for formulation comparison and stability studies).
We thank the reviewer for this essential recommendation. We have prepared a finished-product specification for the 500 mg Prussian blue tablets and incorporated it into the submission. To keep the main text focused (the manuscript is already lengthy), the full specification is provided as Supplementary Table S2 (PB tablets 500 mg - Specifications). The specification is applied throughout the work: it is cited in the formulation comparison and in the stability section when reporting compliance. For transparency, we also point to the corresponding Methods subsections describing each test. Supplementary Table S2: PB tablets (500 mg). Finished product specifications (includes Description, Assay, Uniformity of dosage units per Ph. Eur. 2.9.40, Mean mass per RFE 2.9.5, Hardness, Friability 2.9.7, Disintegration 2.9.1, % Weight loss by TGA, Cs-binding capacity).
Methods (Section 3.8.2): procedures clarified/added for mean mass, hardness, friability, disintegration.
Results/Discussion: explicit references to Spec (Table S2) when stating compliance at release and on stability.
Order of the paper is wrong – materials and methods are listed after results and discussion. Please ensure that there is a clear, logical and easy to follow line of thought throughout the article.
We thank the reviewer for this helpful observation. Our current section order follows the Pharmaceuticals Instructions for Authors and template, which place Materials and Methods after Results and Discussion. We fully agree that presenting Methods before Results can aid readability; however, we adhered to the journal’s recommended structure.
Tablet evaluation is incomplete – stability results should be tested for disintegration, since you stated this as a critical parameter. As a matter of fact, the best approach would be to set up a specification for the product and test the product (all variants) according to the specification, which is also used for stability testing.
We are grateful for this valuable observation. Following the reviewer’s recommendation in order to show a complete we have:
- Set up a full finished-product specification for the 500 mg PB tablets (see Supplementary Table S2), covering all critical quality attributes (description, mean weight, hardness, friability, disintegration, % weight loss by TGA, Cs-binding capacity and assay of PB content).
- Expanded the stability evaluation: the data for disintegration and friability, together with all other specification parameters, have now been incorporated into Tables 8 (accelerated conditions) and 9 (long-term Zone II conditions) of the Results section.
- Added a validated assay of PB content in response to the reviewer’s suggestion. Assay testing has been performed for available stability samples, results are currently available to 18 months, which is the present time-point reached in the ongoing long-term ICH (60-month) stability study. Additional assay data will be generated and reported as the study progresses.
- All stability time-points evaluated to date meet the acceptance criteria of the specification. The revised text in Section 2.8 (Stability results) highlights these updates and cites the expanded Tables 8 and 9.
Changes in the manuscript:
- Tables 8 and 9: now include disintegration time, friability and assay results alongside the other specification parameters.
- Section 2.8 (Results): updated to comment on the full compliance of the tablets with the complete specification and to note that assay data are presently available for 18 months while the ICH long-term study continues.
Tablet hardness is one critical parameter, which is missing from the study – this should be included in the study/specification.
We thank the reviewer for emphasizing this key attribute. Tablet hardness (N) has now been reported in Table 6 (Tablet analysis) and in the stability dataset (Table 9). The test procedure and equipment are described in Section 3.8.2 (Methods). We have also reflected hardness in the Specifications bulletin (as an informative mechanical attribute supporting robustness during handling, packaging and transport).
Section 3.8.2 (Methods): hardness procedure added.
Table 6 / Table 9: hardness values inserted.
Specifications bulletin hardness added as an informative attribute.
Friability is one critical parameter, which is missing from the study – this should be included in the study/specification.
We are grateful for this observation. Friability testing (Ph. Eur./RFE 2.9.7) has been reported in Table 6 (Tablet analysis) and Table 9 (Stability). The test conditions are described in Section 3.8.2 (Methods). The Specifications bulletin now explicitly includes Friability: NMT 1.0% as the release criterion..
Section 3.8.2 (Methods): friability procedure added.
Table 6 / Table 9: friability values inserted.
Specifications bulletin: Friability: NMT 1.0% added as a release limit.
Detailed comments:
Page 3, line 5: The formulation (qualitative composition) of Radiogase is openly available (given in the SmpC). Quantitative composition and exact manufacturing procedure is not openly available. Please correct also all following statements (e.g. page 3 paragraph 3)
Thank you for this clarification. We agree and have revised the text wherever needed to accurately state that the qualitative composition of Radiogardase® is disclosed in the SmPC, whereas its quantitative composition and the exact manufacturing process are not publicly available. We have corrected the statements on page 3 (first and third paragraphs) and harmonized the wording throughout the manuscript.
Page 3, paragraph 1: sentence starting “Currently, the only available pharmaceutical product…” revised.
Page 3, paragraph 3: sentence starting “Owing to the critical importance of antidotes and the lack of a medication with PB that is globally accessible…” revised.
Aim paragraph (end of page 3): wording adjusted to emphasize the open-access tablet formulation and process description, not implying that Radiogardase® lacks qualitative disclosure.
Page3: “the elimination of the dependency on external suppliers,« this statement is not true, since external suppliers for API and excipients are needed. In the next sentences you explain that »minimizes potential supply chain issues in crisis situations« and contradicts your previous statement.
We appreciate this observation and agree. Our intention was to emphasize that direct compression reduces reliance on specialized components (hard-gelatin capsule shells), not that external suppliers are no longer required. We have revised the text to state that API and excipients continue to be sourced from qualified external suppliers, while the tableting process shortens the production cycle and reduces dependence on specific components, thereby mitigating (not eliminating) supply-chain risks.
Changes in manuscript:
Page 3, paragraph beginning “A primary goal of this project…”, sentence about “elimination of the dependency on external suppliers” rewritten and paragraph harmonized.
Page 3, objective paragraph adjusted for consistency.
Page 3: sentence “In addition, the stability of the tablets increased under humid conditions, allowing for simplified formulations.” Is not supported by the data in the article.
We agree. The sentence could be read as a data-based claim specific to our study, which we did not substantiate. We have removed this statement to avoid overinterpretation. The paragraph now focuses on operational advantages of direct compression (shorter production cycle, reduced reliance on specialized components) and on supply-chain considerations, which are supported by our methods and rationale.
Page 3, paragraph beginning “A primary goal of this project…”, sentence about stability under humid conditions deleted and paragraph reworded for clarity and style.
The word “agglutinant« should be replaced by »binder« for clarity reasons, since the latter word in commonly used.
We have replaced the term “agglutinant” with “binder” throughout the manuscript.
Correction made whenever the word appeared on the manuscript
Page 5, paragraph 2 and 3: Mg stearate and glyceryl behenate are not the most common glidants, but lubricants. There is a mixture of both words in the paragraphs – glidant and lubricant are not exact synonyms – glidants improve flow properties while lubricants reduce friction and sticking. Please correct and clarify. If glidant in the correct word, why is there not mention of colloidal silicon dioxide, the most common glidant? From the paragraphs I conclude that the API had adequate flowability and no glidant is required, but lubricant is usually required and behenate was selected over stearate.
We appreciate this detailed observation and agree with the corrections.
- We have corrected the classification of magnesium stearate and glyceryl behenate from “glidants” to “lubricants” and clarified their specific function (to reduce die-wall friction and sticking during compression).
- We clarified that glyceryl behenate was selected as the lubricant instead of magnesium stearate, citing its favorable performance and to avoid potential gastrointestinal effects occasionally reported for magnesium stearate.
These corrections have been made in Page 5, paragraphs 2 and 3.
Page 5, paragraph above heading 2.2 seems to be out of place. Maybe a leftover of revision or AI instructions?
We sincerely thank the reviewer for carefully spotting this issue. The paragraph above heading 2.2 was an unintended remnant from the journal template and had no relevance to the manuscript content. We have removed this paragraph in the revised version to ensure the section flows correctly.Page 5: deleted the stray template paragraph that appeared immediately above heading 2.2.
2.2. Developed Formulations – the title is misleading tested or evaluated formulations might be better.
We thank the reviewer for this helpful suggestion. We agree that the section title “Developed Formulations” could be interpreted as referring only to the design stage, while the content actually describes the formulations that were tested and evaluated. Accordingly, we have retitled the section to “Tested Formulations” in the revised manuscript to better reflect its content.
Section heading 2.2: changed from “Developed Formulations” to “Tested Formulations”.
DSC thermograms – the interpretation is lacking. Thermograms of pure substances could also be provided for better interpretation. For example: MCC is described as having a characteristic peak at 50-150 °C. No such peak is observed in the thermograms. For mannitol, the peaks described are probably reversed as described, since the shape of PB peak is different on all other thermograms if your interpretation is correct. Substantial revision is needed.
We sincerely thank the reviewer for these valuable observations regarding the DSC analysis.
In response, we repeated the DSC measurements for all binary PB-excipient mixtures and additionally recorded the thermograms of each pure excipient and of PB alone. These new data are now included as Supplementary Figure S1, and Figure 2 has been updated with the repeated binary runs. The Results section (2.3) has been thoroughly revised to:
- Identify the broad 50-120 °C endotherm of microcrystalline cellulose (bound-water loss), which is evident in the pure material (Figure S1) but masked in the PB-MCC mixture by the stronger PB endotherm.
- Clarify that in the PB-mannitol binary mixture the characteristic mannitol melting endotherm (165-169 °C) and the PB dehydration endotherm (~155 °C) overlap and appear as a single broadened thermal event, as confirmed by comparison with the pure mannitol thermogram.
- Report that, in the repeated PB–glyceryl behenate run, the PB endotherm (~155-200 °C) is clearly present together with the lipid-melting peak (70-85 °C), with only a slight downward shift attributable to sample packing.
- Highlight that in the PB-magnesium stearate blend the PB endotherm appears broadened and partially degraded, suggesting a possible interaction which, together with the poor flow properties, justifies excluding magnesium stearate from the final DC5 formulation.
Page 17: more details are needed for the wet granulation (speed of kneading, speed of mill..)
We thank the reviewer for this valuable observation.
The Methods section has been updated to include the missing operating parameters: kneading speed (60 rpm) in the double-Z kneader, rate of water addition (≈10 % of the total water volume per minute), granulator type and speed (oscillating granulator, 40 rpm, 1 mm mesh) and drying conditions (tray oven at 90 °C, two-shelf configuration, shallow bed, moisture monitored until ≤ 1 % w/w).
Methods – Wet granulation section (page 17 section3 .5.1) revised and replaced.
Chapter 3.6 Screening – what was the yield of the process? It would be beneficial to add yield to all methods and formulations tested, since yield is also an important factor in table manufacturing..
We thank the reviewer for this practical suggestion. We have now added yield data using consistent definitions across manufacturing routes. Specifically:
A new Methods subsection (Section 3.9 – Yield definitions and calculations) describes how yields were defined and computed, Overall tablet yields for each route are reported in the Results & Discussion (Section 2.5, page 12). Step yields and mass balances are provided in Supplementary Table S1.
Note: As DC1–DC4 were exploratory screening formulations that were discontinued for not meeting CQAs, process yields were not recorded for those batches.
Changes in manuscript:
Section 3.9 (Methods): new subsection Yield definitions and calculations.
Section 2.5 (Results & Discussion, p. 12): overall yields added (textual summary).
Supplementary Table S1: step yields and overall yield summary (DC5 development & pilot, WG, DG).
Chapter 3.7. Tablet Manufacturing – why were all tablets prepared at the same compression force? What is the rationale behind this? Please explain.
We thank the reviewer for this helpful comment. On this tablet press, the compression setting is controlled by an analog, notched lever in 0.5-unit increments (“Bonals units,” i.e., instrument units rather than absolute force in kN). For each batch, the setting was tuned in situ to the highest feasible notched position, which falls within an operating band of 5.0–6.0 units (nominal target ≈ 5.5) that allowed stable continuous operation at 30 rpm with the selected tooling. This approach ensured comparable compaction conditions across formulations, given the instrument’s discrete control. We have clarified this point in Section 3.7. We also acknowledge that some formulations exhibited defects (as explained on the manuscript), which we attribute to a combination of formulation/granulation characteristics and the press’s capacity constraints.
Section 3.7. Tablet Manufacturing – final sentences revised.
Reviewer 2 Report
Comments and Suggestions for Authors
The article addresses a definite need to have publicly available description of manufacturing process for Prussian blue tablets. However, it needs some improvement in order to deliver the content in its entirety.
Firstly, the organization of the articles is slightly confusing (methods after the results) and a great deal of attention is given to some details (e.g. DSC and SeDeM), while other parts are lacking. The article needs to be reorganized and shortened to be clearer.
In tablet manufacturing, the specification of tablets is the single most useful and critical document that defines and ensures repeatable quality and clinical efficacy of the tablets. Specification for your tablets should be provided and relied upon in the article (for formulation comparison and stability studies)
Order of the paper is wrong – materials and methods are listed after results and discussion. Please ensure that there is a clear, logical and easy to follow line of thought throughout the article.
Tablet evaluation is incomplete – stability results should be tested for disintegration, since you stated this as a critical parameter. As a matter of fact, the best approach would be to set up a specification for the product and test the product (all variants) according to the specification, which is also used for stability testing.
Tablet hardness is one critical parameter, which is missing from the study – this should be included in the study/specification.
Friability is one critical parameter, which is missing from the study – this should be included in the study/specification.
Detailed comments:
Page 3, line 5: The formulation (qualitative composition) of Radiogase is openly available (given in the SmpC). Quantitative composition and exact manufacturing procedure is not openly available. Please correct also all following statements (e.g. page 3 paragraph 3)
Page3: “the elimination of the dependency on external suppliers,« this statement is not true, since external suppliers for API and excipients are needed. In the next sentences you explain that »minimizes potential supply chain issues in crisis situations« and contradicts your previous statement.
Page 3: sentence “In addition, the stability of the tablets increased under humid conditions, allowing for simplified formulations.” Is not supported by the data in the article.
The word “agglutinant« should be replaced by »binder« for clarity reasons, since the latter word in commonly used.
Page 5, paragraph 2 and 3: Mg stearate and glyceryl behenate are not the most common glidants, but lubricants. There is a mixture of both words in the paragraphs – glidant and lubricant are not exact synonyms – glidants improve flow properties while lubricants reduce friction and sticking. Please correct and clarify. If glidant in the correct word, why is there not mention of colloidal silicon dioxide, the most common glidant? From the paragraphs I conclude that the API had adequate flowability and no glidant is required, but lubricant is usually required and behenate was selected over stearate.
Page 5, paragraph above heading 2.2 seems to be out of place. Maybe a leftover of revision or AI instructions?
2.2. Developed Formulations – the title is misleading tested or evaluated formulations might be better.
DSC thermograms – the interpretation is lacking. Thermograms of pure substances could also be provided for better interpretation. For example: MCC is described as having a characteristic peak at 50-150 °C. No such peak is observed in the thermograms. For mannitol, the peaks described are probably reversed as described, since the shape of PB peak is different on all other thermograms if your interpretation is correct. Substantial revision is needed.
Page 17: more details are needed for the wet granulation (speed of kneading, speed of mill..)
Chapter 3.6 Screening – what was the yield of the process? It would be beneficial to add yield to all methods and formulations tested, since yield is also an important factor in table manufacturing.
Chapter 3.7. Tablet Manufacturing – why were all tablets prepared at the same compression force? What is the rationale behind this? Please explain.
Comments on the Quality of English LanguageGrammatical and spelling errors, some language errors (comma placemen, “In the tablet is used mainly as an«, »To ensure demanding flow properties, »….) - a revision by a native speaker is required.
Author Response

(The authors gave the same response as above.)

Round 2
Reviewer 1 Report
Comments and Suggestions for Authors
Dear Author,
Your revised manuscript entitled “Innovative Galenic Formulation of Prussian Blue Tablets: Advancing Pharmaceutical Applications” has been carefully re-evaluated. The revisions undertaken have addressed the earlier concerns in a thorough and satisfactory manner. The manuscript now presents a clear, well-structured, and comprehensive account of the formulation development, manufacturability, and stability of Prussian Blue tablets. The additional literature context and detailed physicochemical characterizations further enhance the novelty and relevance of the study. Overall, the responses and revisions are considered successful, and the manuscript can be regarded as suitable for publication in its present form.
Kind regards,
Author Response
We are grateful for your comments and the improvements they have brought to the final quality of the article.
Reviewer 2 Report
Comments and Suggestions for Authors
The article is substantially improved, the readability is much better, and the logic is easier to follow.
However, some important information is still missing (addressed in the comments).
When this data is added, I believe the article will be ready for publication.
Language still needs to be improved.
Comments:
Chapter 2.3 DSC – Only Fig 2 is commented in the first sentence/first paragraph. Since the DSC thermogram of individual components were also recorded, they should be referenced in a sentence and indicated that they are a part of the supplement. Introducing reference to Fig S1 is not that clear.
Chapter 2.5.1. – The first paragraph should be corrected to reflect new addition to the table 6 (Hardness and friability). Further, the significance of these two parameters should be discussed in the paragraph, since they are not mentioned.
Line 392: It should be mentioned that Table S1 is in the supplement for clarity reasons.
Line 435 it should be mentioned that Table S2 is in the supplement, otherwise the reader might be confused.
Is it possible to include the data for Radiogardase in Table 6? The data would be most helpful.
Figure 5: legend is missing
Table 8: Friability data is missing for the storage conditions. It is referenced in the text, however, so it probably exists.
Chapter 3.5.1. – The amount of total added water should be mentioned here, since it is not provided in the composition.
Comments on the Quality of English LanguageGrammatically incorrect of logically misleading sentences are still present - a revision by a native speaker is required.
For instance: »To ensure demanding flow properties» is incorrect, since you want flow properties to be undemanding (i.e. simple) – probably “To ensure demanded flow properties” or even better “To ensure required flow properties” is the true meaning that you wanted to express.
Another example: " The mixture, in which PB represents a 71.43% of each formulation, has a good base flowability because of the PB good flowability." Is grammatically poor and a bit misleading (“71.43% of each formulation “is unclear) – “The mixture, in which PB represents a 71.43% of the formulation has a good flowability because the flowability of PB is good in itself. « is much better and clearer.
There are many more such examples.
Author Response
The article is substantially improved, the readability is much better, and the logic is easier to follow.
However, some important information is still missing (addressed in the comments).
When this data is added, I believe the article will be ready for publication.
Language still needs to be improved.
Comments:
Chapter 2.3 DSC – Only Fig 2 is commented in the first sentence/first paragraph. Since the DSC thermogram of individual components were also recorded, they should be referenced in a sentence and indicated that they are a part of the supplement. Introducing reference to Fig S1 is not that clear.
We thank the reviewer for this helpful observation. We have revised the first paragraph of Section 2.3 to explicitly state that the DSC thermograms of the individual components are provided in the Supplementary Materials and to clarify their relationship to Figure 2. The following sentence was added to the paragraph (Section 2.3, first paragraph, line 231):
“For completeness, DSC thermograms of the individual components (PB and each excipient) were also acquired under identical conditions; the corresponding reference profiles are provided in the Supplementary Materials (Figure S1) and are cited, where pertinent, to support the assignment of thermal events observed in the PB-excipient binary systems shown in Figure 2.”
Chapter 2.5.1. – The first paragraph should be corrected to reflect new addition to the table 6 (Hardness and friability). Further, the significance of these two parameters should be discussed in the paragraph, since they are not mentioned.
We thank the reviewer for the suggestion. We revised Section 2.5.1 to incorporate and succinctly interpret the hardness and friability results (Table 6), clarifying their role in the formulation rationale and in the selection of the final direct-compression blend. We also state the intended balance between mechanical integrity and very rapid disintegration and add a brief comparison of the granulated formulations (WG and DG) to align the discussion with these performance criteria, with references harmonized to the Supplementary Materials.
Line 392: It should be mentioned that Table S1 is in the supplement for clarity reasons.
We agree and thank the reviewer for the helpful comment. We revised the sentence to explicitly reference the Supplementary Materials, now reading “(Table S1, provided in the Supplementary Materials)” for clarity and consistency.
Line 435 it should be mentioned that Table S2 is in the supplement, otherwise the reader might be confused.
We thank the reviewer for the suggestion. In consequence, we added an explicit reference to the Supplementary Materials at line 435 (“Table S2, provided in the Supplementary Materials”). In addition, we harmonized the manuscript so that all mentions of supplementary tables/figures now consistently read “provided in the Supplementary Materials
Is it possible to include the data for Radiogardase in Table 6? The data would be most helpful.
We thank the reviewer for this suggestion. We have added Radiogardase® as a comparator in Table 6. In line with pharmacopoeial scope, tablet-specific mechanical tests (hardness and friability) are not applicable to hard capsules; therefore, they are reported as N/A, with an explanatory footnote. To enable comparison, the mean unit weight of Radiogardase® (capsule content) was determined in accordance with RFE 2.9.5, and the discussion in Section 3.8.2 now notes the shorter disintegration achieved by our formulations relative to Radiogardase®. Additionally, the manuscript now specifies the comparator as Radiogardase®-Cs 500 mg hard capsules (batch no. 24002125).
Figure 5: legend is missing
We thank the reviewer for the observation. We have re-checked the manuscript and confirm that the legend for Figure 5 is present and reads:
“Figure 5. Mean percentage of PB dispersed (± SD, n = 6) over time in simulated gastric fluid without enzymes (pH 1.2) at 37 ± 0.5 °C, using the Real Farmacopea Española (RFE) Apparatus 1 (equivalent to USP paddle method) at 100 rpm.”
During this verification, we noted that the legend for Figure S1 in the Supplementary Materials was not included; this has now been added as:
“Figure S1. DSC thermograms of the PB raw material, all single-excipient samples, and the direct-compression blend DC5.”
We have also reviewed the remaining figures to ensure captions are consistently present and formatted.
Table 8: Friability data is missing for the storage conditions. It is referenced in the text, however, so it probably exists.
We thank the reviewer for noticing the missing information. The friability results under the specified storage conditions have now been added to Table 8.
Chapter 3.5.1. The amount of total added water should be mentioned here, since it is not provided in the composition.
We thank the reviewer for the remark. The manuscript already stated in Section 2.2 that wet granulation required 30% (w/w) Milli-Q water. For clarity, we have now added an explicit sentence in Section 3.5.1 specifying the total water added for wet granulation; for the 1.00 kg development batch, this corresponds to 300 g of water. This aligns Sections 2.2 and 3.5.1 and clarifies the processing conditions.
Comments on the Quality of English Language
Grammatically incorrect of logically misleading sentences are still present - a revision by a native speaker is required.
For instance: »To ensure demanding flow properties» is incorrect, since you want flow properties to be undemanding (i.e. simple) – probably “To ensure demanded flow properties” or even better “To ensure required flow properties” is the true meaning that you wanted to express.
Another example: " The mixture, in which PB represents a 71.43% of each formulation, has a good base flowability because of the PB good flowability." Is grammatically poor and a bit misleading (“71.43% of each formulation “is unclear) – “The mixture, in which PB represents a 71.43% of the formulation has a good flowability because the flowability of PB is good in itself. « is much better and clearer.
There are many more such examples.
We appreciate this observation and have performed a comprehensive, line-by-line language revision of the manuscript to improve clarity, grammar, and readability, adopting consistent US English throughout. We corrected awkward or ambiguous constructions, standardized scientific terminology, and harmonized punctuation, units, and symbols (e.g., percent signs, ranges, °C, ICP–OES nomenclature).
Representative corrections include:
Original: “To ensure demanding flow properties.”
Revised: “To ensure the required flow properties.”
Original: “The mixture, in which PB represents a 71.43% of each formulation, has a good base flowability because of the PB good flowability.”
Revised: “In the mixture, PB constitutes 71.43% of the formulation; consequently, the blend exhibits good flowability, as PB itself flows well.”
Global edits applied across the text, tables, and figure legends include (non-exhaustive list):
Standardized US spelling (e.g., behavior, color, aluminum, pharmacopeial, caliper).
Consistent style for percentages and units (no extra spaces; 5, 15, 30, and 60 min; 37 ± 0.5 °C), and en dashes for numeric ranges (120–160 °C).
Consistent comparative usage (compared with), subject–verb agreement, and article use.
Clarified instrument names and acronyms (e.g., ICP–OES), and polished figure/table captions (including the addition/clarification of Figure S1 legend).
Simplification and clarification of expressions: replaced verbose or tautological constructions with concise, precise wording; resolved ambiguous references; standardized transitional phrases (e.g., “therefore,” “thus,” “in addition”); reduced passive voice where it improved clarity; and removed redundancies.
We believe these revisions address the reviewer’s concern and materially improve the manuscript’s readability and precision. We would be happy to consider any further specific instances the reviewer may wish to flag.
Round 3
Reviewer 2 Report
Comments and Suggestions for Authors
The manuscript has improved; however, it is hard to read due to many correction. Next time, a version without the corrections should also be provided.
Language has improved and is acceptable, but sometimes the clarity is still lacking.
Some minor corrections:
Line 386
“Moreover, the wetting step with water is safe for the PB raw material because” should be rewritten as “Moreover, the wetting step with water is not affecting the PB raw material because”
Line 392
“Wet granulation leads to more uniform and spherical shapes, which flow better and are more compactable; however, dry granulation produces irregularly shaped granules,” should be rewritten as “Wet granulation leads to more uniform and spherical shapes, which flow better and are more compactable; dry granulation produces irregularly shaped granules,”
Line 421
“The insufficient flowability of magnesium stearate resulted in worse die filling during the compression” should be rewritten as “The insufficient flowability of magnesium stearate resulted in insufficient die filling during the compression”
Figure 5: legend is missing – not the caption/description of the Figure, but the explanation which line (blue or orange) is your and which is the reference product.
Table 8 and 9: the average results for three batches are presented. Standard deviations should also be given or at least commented in the text.
Author Response
The answers are enclosed in the attached file.
